# Characterizing clouds with the CCClim dataset, a machine learning cloud class climatology

Arndt Kaps[1], Axel Lauer[1], Rémi Kazeroni[1], Martin Stengel[2], and Veronika Eyring[1,3]

[1]Deutsches Zentrum für Luft- und Raumfahrt e.V. (DLR), Institut für Physik der Atmosphäre, Oberpfaffenhofen, Germany
[2]Deutscher Wetterdienst (DWD), Offenbach, Germany
[3]University of Bremen, Institute of Environmental Physics (IUP), Bremen, Germany

**Correspondence:** Arndt Kaps (arndt.kaps@dlr.de)

**Abstract.** We present a new Cloud Class Climatology dataset (CCClim), quantifying the global distribution of established morphological cloud types over 35 years. CCClim combines active and passive sensor data with machine learning (ML) and provides a new opportunity for improving the understanding of clouds and their related processes. CCClim is based on cloud property retrievals from the European Space Agency's (ESA) Cloud_cci dataset, adding relative occurrences of eight major cloud types, designed to be similar to those defined by the World Meteorological Organization (WMO) at $1°$ resolution. The ML framework used to obtain the cloud types is trained on data from multiple satellites in the Afternoon Constellation (A-Train). Using multiple spaceborne sensors reduces the impact of single-sensor problems like the difficulty of passive sensors to detect thin cirrus or the small footprint of active sensors. We leverage this to generate sufficient labeled data to train supervised ML models. CCClim's global coverage being almost gapless from 1982 to 2016 allows for performing process-oriented analyses of clouds on a climatological time scale. Similarly, the moderate spatial and temporal resolutions make it a lightweight dataset while enabling straightforward comparison to climate models. CCClim creates multiple opportunities to study clouds, of which we sketch out a few examples. Along with the cloud type frequencies, CCClim contains the cloud properties used as inputs to the ML framework, such that all cloud types can be associated with relevant physical quantities. CCClim can also be combined with other datasets such as reanalysis data to assess the dynamical regime favoring the occurrence of a specific cloud type in association with its properties. Additionally, we show an example of how to evaluate a global climate model by comparing CCClim with cloud types obtained by applying the same ML method used to create CCClim to output from the icosahedral nonhydrostatic atmosphere model (ICON-A).

CCClim can be accessed via the digital object identifier: 10.5281/zenodo.8369202 (Kaps et al., 2023b)

## 1 Introduction

An improved understanding of the interactions of clouds within the climate system is integral for making more robust projections of future climates. Observational data inform on cloud states in the current climate and are therefore a crucial part of this process (Zelinka et al., 2020). While observations continue to be taken at the surface or from aircraft, the global or near-global datasets that can be obtained from spaceborne remote sensing instruments play an important role in climate science. Satellite observations have been used to better understand the distribution of clouds on the global scale and to assess the quality of cloud

representations in global climate models (Vignesh et al., 2020; Ceppi et al., 2016; Wall et al., 2022; Bodas-Salcedo et al., 2012; Lauer et al., 2023). However, uncertainties and differences among available satellite products make it difficult to interpret the data objectively and to evaluate climate model results (Evan et al., 2007; Zhang, 2005; Dubovik et al., 2021). Climate science would therefore significantly benefit from satellite products that are more accurate over large areas as well as easily comparable and interpretable. In creating a new dataset, we address these challenges by combining active and passive sensor data, providing a global and long-term time series of relative cloud-type amounts.

Current satellite products typically include physical variables retrieved from the measured radiation such as cloud top temperature, cloud water path or thermodynamic phase. These retrievals are, however, subject to sensor limitations. Passive sensors can typically only measure integrated properties of the complete atmospheric column and characterize the topmost cloud layer. For this reason, only limited insight may be gained into overlapping clouds with passive sensors. Optically thin clouds or clouds over snowy landscapes are also challenging for passive satellite instruments due to limited measurement contrast relative to the surface. In contrast, active instruments such as lidar and radar sensors can resolve certain properties vertically but provide small measurement footprints limited to a narrow swath, such that global coverage with short revisit periods is usually impossible. A common approach is, therefore, to combine measurements from multiple sensors to obtain a more complete assessment of the atmosphere (Haynes et al., 2011; Jiang et al., 2012; Wang et al., 2016; Stubenrauch et al., 2017). Additionally, measurements can be subject to calibration issues, which may introduce further deviation between individual instruments and/or simulations (Loeb et al., 2009). With improving machine learning (ML) capabilities, the opportunity arises to produce smart combinations of measurements from active and passive sensors to provide enhanced observational products benefiting from both types of sensors (Reichstein et al., 2019).

Complementing existing products, we present a new dataset for the climatology of cloud types, named CCClim (Kaps et al., 2023b), produced using ML algorithms trained with a combination of data from active and passive instruments. The cloud types contained in CCClim are physically consistent with most of the major cloud genera defined by the World Meteorological Organisation (WMO) making the dataset easier to compare to human observations than other definitions as used e.g. in Kurihana et al. (2022). A consistent, long-term cloud-type dataset like CCClim is required to empirically study the processes governing the interaction of clouds within the climate system, which in turn is a necessary component in improving current climate models (Li et al., 2015).

Climatologies of the cloud type from remote sensing data have long been in use to categorize multi-dimensional large-scale cloud properties, like in the dataset created by the International Satellite Cloud Climatology Project (ISCCP) (Schiffer and Rossow, 1983). Now, with increasingly advanced measurements and computational capabilities such categorization has started incorporating other quantities beyond cloud top pressure (*ptop*) and optical depth (*cod*) as in Rossow and Schiffer (1999). More recent versions of the ISCCP dataset have been used in combination with ML methods to provide data categorized by cloud regime (e.g. Young et al., 2018; Tselioudis et al., 2013; Tzallas et al., 2022). These regimes are defined by unsupervised clustering algorithms on observational data of horizontal resolutions in the order of tens to hundreds of kilometers (Tselioudis et al., 2021). Using unsupervised methods is a popular strategy when labeled data are sparse or expensive to produce. However, clustering strategies, especially when based on low-resolution data, can struggle to produce meaningful cloud regimes, as

such strategies are prone to compensating errors and uncertainties introduced by overlapping clouds (McDonald and Parsons, 2018). Creating manually cloud-labeled satellite data of sufficient quantity is possible, but comes with considerable cost and limitations (Stevens et al., 2019; Rasp et al., 2020). In contrast, Zantedeschi et al. (2019) used sparse WMO-like labels obtained from active instruments aboard CloudSat (Stephens et al., 2002) and the Cloud-Aerosol Lidar and Infrared Pathfinder Satellite Observation (CALIPSO) (Winker et al., 2003) to train a convolutional neural network (CNN) in a supervised framework to label images from the passive Moderate Resolution Imaging Spectroradiometer (MODIS) (Platnick et al., 2003, 2017; Stephens et al., 2018; Zantedeschi et al., 2019). Another way of training a supervised model was employed by Kuma et al. (2023), who combined measurements from the Clouds and the Earth's Radiant Energy System (CERES) with cloud types observed from ground stations to create a labeled dataset. Using a supervised method with prescribed classes can arguably be restrictive, which is why some studies deliberately employ unsupervised methods to find more distinct classes. Kurihana et al. (2021) combined an autoencoder network with a clustering algorithm, finding twelve cloud-type clusters from $128 \times 128$ pixel patches of MODIS data. They used a refined version of this method to produce a dataset of 42 individual cloud-patch classes (Kurihana et al., 2022).

The CCClim dataset presented here has been created using a combination of supervised ML methods trained using passive as well as active satellite sensors, using a previously published framework (Kaps et al., 2023a).

We provide an overview of the datasets and methods used in training, application and evaluation in Section 2. Section 3 deals with the contents of CCClim and shows examples of how it can be used to study clouds. The potential of using CCClim to evaluate global climate models is indicated in Section 4. Finally, we discuss the scope of the dataset's capabilities in Section 5.

## 2 Data and Methods

We trained two ML models that are applied consecutively to predict each cloud type's relative frequency of occurrence (RFO) in low-resolution grid cells similar to the resolution of most current global climate models. The reasoning behind our approach is explained in section 2.3. The first stage is trained on the CUMULO dataset (year 2008) created by Zantedeschi et al. (2019). CUMULO contains physical variables (see Table 2) obtained from the MODIS Cloud Product MYD06 dataset from the Aqua satellite, which we use as input features (Platnick et al., 2003, 2017). As Aqua is part of the A-Train constellation, its measurements can be aligned with measurements from other A-Train satellites, such as CloudSat and CALIPSO (Stephens et al., 2018). These are used in CUMULO to provide target labels as WMO-like cloud types from CloudSat's 2B-CLDCLASS-LIDAR dataset (hereafter CC-L) (Wang, 2019). We call the cloud types "WMO-like" because they are defined to correspond to eight of the ten WMO genera, with the Dc (deep convective) type replacing the WMO's cumulonimbus and cirrocumulus/cirrostratus being contained in the Ci (cirrus) type. While these classes are defined to be consistent with the WMO definitions, misclassifications can occur, caused for example by the small footprint of the active sensors. In CUMULO, coinciding measurements from MODIS and CC-L are aligned at the pixel level. The second stage is a regression model trained on coarse-grained output from the first stage. Finally, CCClim is produced by applying the trained regression model to the European Space Agency (ESA) Cloud_cci L3U-AVHRR-PM dataset (1982-2016) (Stengel et al., 2019, 2020), in the following called ESACCI. Application

of an ML model to a different dataset (out-of-distribution) can be problematic and great care has to be taken to ensure no unexpected errors occur. Out-of-distribution application of ML models has been done in climate science before(e.g Yuval and O' Gorman, 2020; Kuma et al., 2023; Wang, 2019). Here, it is important to estimate the uncertainties induced by the domain shift. In our case, the regression model trained on MODIS cloud properties is applied to similar AVHRR retrievals. Applicability could be demonstrated by the reasonable reproduction of the geographical distribution of all cloud types when applying the model to ESACCI data (AVHRR), also providing an uncertainty estimate as documented in Kaps et al. (2023a). This suggests that the method is robust enough to be applied to different datasets if they represent similar basic physical properties.

## 2.1 Data

### 2.1.1 ESA Cloud_cci L3U-AVHRR-PM (ESACCI)

ESACCI contains twice daily measurements from the Advanced Very High Resolution Radiometer (AVHRR) on a $0.05°$-grid (L3U data). It contains a comprehensive set of cloud and radiative flux products (see Table 1 in Stengel et al. (2020)), including cloud water path and effective radius measurements together with a flag for the cloud top thermodynamic phase. We use this flag to assign a single phase to the cloud column, which is - while being a considerable simplification - an effective way to separate cloud water path into *lwp*/*iwp* and effective radius into *cerl*/*ceri* for liquid and ice clouds, respectively. This separation is required to replace categorical with continuous variables for our machine learning methods. For cases with an unclear cloud top phase, we assign half of the *cwp* value to *iwp* and *lwp*, each.

### 2.1.2 MYD06 (MODIS Cloud Product)

MODIS is a passive sensor aboard the Aqua satellite, which is polar-orbiting and sun-synchronous, providing measurements at near-constant local time. The MODIS Cloud Product is a level 2 product, containing retrievals of physical variables (see Table 2). The MODIS data are used in CUMULO to provide global coverage at daily resolution. Some of the MODIS retrievals are only available during daytime, which is why CUMULO only contains daytime data. The physical variables are provided as $1 \times 1 \, \mathrm{km}^2$ resolution multivariate images of $1354 \times 2030$ pixels. CUMULO contains Version 6.1 of the MODIS Cloud Product.

### 2.1.3 2B-CLDCLASS-LIDAR (CC-L)

The CC-L dataset was created by combining measurements from the Cloud Profiling Radar aboard CloudSat and the CALIOP Lidar aboard CALIPSO (Stephens et al., 2002; Winker et al., 2009). CC-L contains WMO-like cloud-type labels obtained by a fuzzy-logic classifier (Wang, 2019), which act as cloud-type labels in CUMULO. The classifier assigns one of eight classes that are designed to be similar to the WMO cloud genera by using characteristics that are typically associated with these genera(Wang, 2019). Eight features, such as cloud height, phase and precipitation are used allowing for a more accurate cloud type assignment than the popular ISCCP classification scheme which only uses two inputs (Rossow and Schiffer, 1999). CC-L can contain up to 10 vertical layers of cloud, but the majority of the data is single-layer. In CUMULO, multi-layer clouds are reduced to a single label by choosing the most common type per column. This is to be more consistent with the 2-dimensional

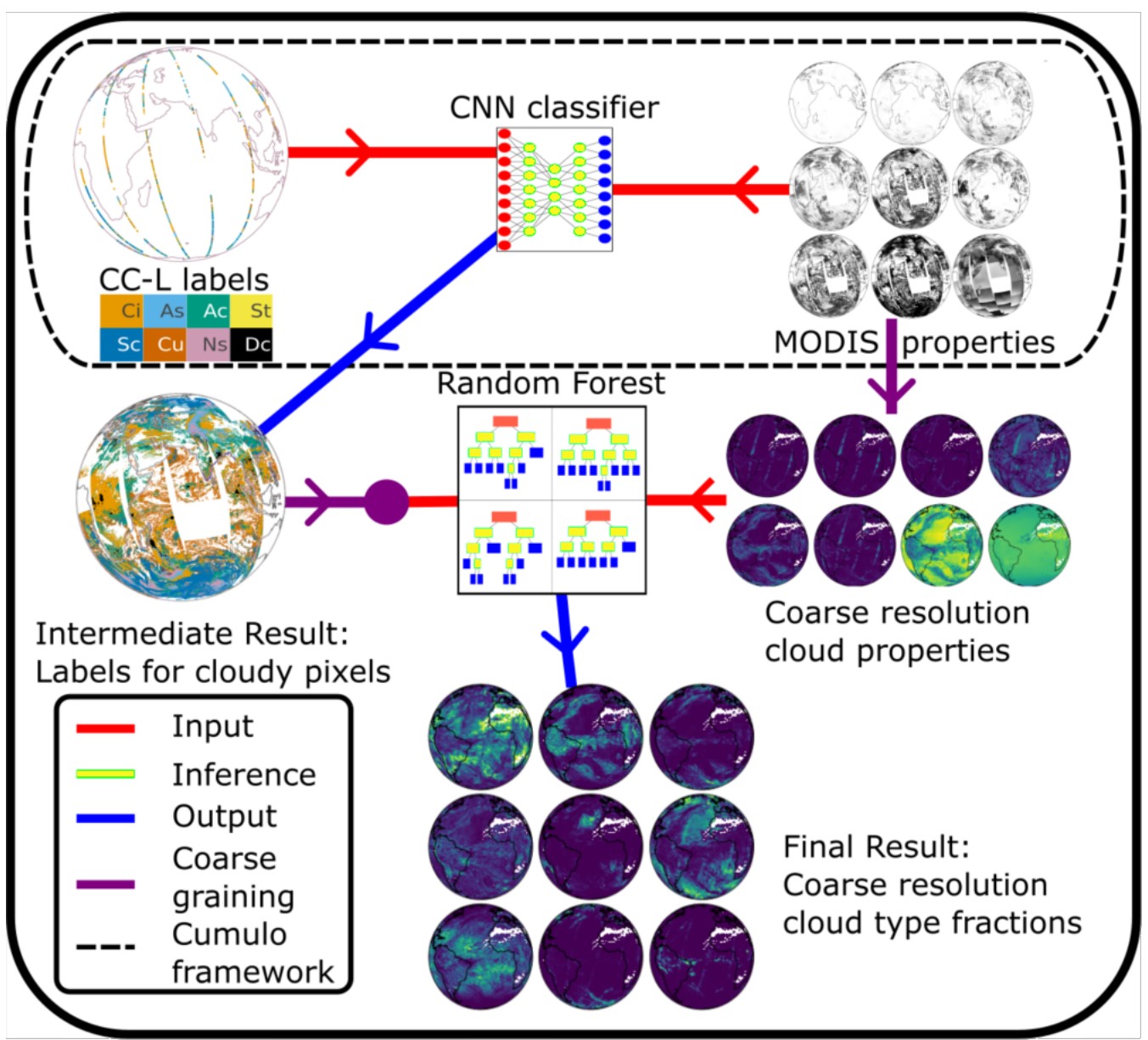

**Figure 1.** Schematic of the training of the two machine learning models. The second stage is trained on coarse-grained output from the first stage. The trained RF is then applied to ESACCI to obtain CCClim.

measurements from MODIS. Because the radar and lidar sensors have a very narrow footprint, there is less than one label per 1354 MODIS pixels, depending on the amount of cloud cover, resulting in a narrow track of labels overlayed on MODIS samples.

### 2.1.4 ERA5 Reanalysis

We use data from the European Centre for Medium-Range Weather Forecasts (ECMWF) fifth-generation reanalysis ERA5 (Hersbach et al., 2020) to assess the plausibility of the derived cloud types in CCClim. Specifically, we use monthly means of the vertical velocity at 500 hPa ($\omega_{500}$) and the sea-surface temperature $SST$ as proxies for the dynamical and local thermodynamic conditions (Bony et al., 2004). These data are available for the complete period covered by CCClim, such that we can spatiotemporally co-locate them.

### 2.2 Method

The first of our two ML models is a pixel-wise classifier based on the Invertible Residual Network framework also used by Zantedeschi et al. (2019) (Behrmann et al., 2019). However, we use the retrieved physical quantities instead of radiances as inputs to our network for better interpretability. The pixel-wise CNN classifier trained to predict CC-L cloud type labels from MODIS observations is used to obtain a label for each cloudy $1 \times 1 \mathrm{km}^2$ pixel in the MODIS data. This process requires two important assumptions: first, that the most common cloud type is representative of the column-integrated cloud properties, and second, that the off-nadir MODIS retrievals are statistically similar to those close to nadir, for which CC-L labels are available. While viewing geometry is part of the retrievals for the cloud optical properties (Platnick et al., 2017), across-track angular dependencies have been identified as a source of uncertainties in the MODIS Collection 5 data (Maddux et al., 2010; Horvath et al., 2014). Similar effects are expected for the Collection 6 data used here, but their magnitude is unclear and no published correction methods are available. These effects are, however, expected to be less relevant for long-term averages (Maddux et al., 2010). Following the data usage in many other studies (e.g. Oreopoulos et al., 2016; Bennartz and Rausch, 2017; Cho et al., 2021) we have therefore opted to make the above assumption instead of introducing new uncertainties by trying to correct these non-trivial effects. The classifier achieves good accuracy for most classes and deals well with the significant class imbalance, except for the classes stratus (St) and stratocumulus (Sc). Both of these cloud types can occur under similar conditions, while Sc is much more frequent than St.

For the second stage, the CUMULO dataset, now fully labeled by our classifier, is coarse-grained to a horizontal resolution of $100 \times 100$ MODIS pixels, comparable with the size of a grid cell of a typical global climate model. The coarse-graining entails averaging the input features and converting the pixel labels into RFOs for each cloud type, i.e. the relative amounts of each type per cell. Pixels without a label are treated as undetermined, such that clear-sky is also included in the "undetermined" class. As some features such as *ptop* are only defined if a cloud is present, we average these only over pixels with a cloud label. We then apply a Random Forest (RF (Breiman, 2001)), which is used as a regression model to predict the RFO of each of the nine classes (see Table 1). In Kaps et al. (2023a), the physical consistency of the predicted RFOs was validated by using the independent ESA Cloud_cci dataset as input to the RF, which showed good agreement with the cloud-type distribution in

CC-L. This shows that while the RF is trained using MODIS data as inputs, it generalizes well to other datasets. Sensitivity tests with the outermost 300 pixels on each side discarded during training of the RF to reduce the viewing angle effects (not shown) show a decreased performance for all of the eight cloud classes as measured by their mean $R^2$-Score and a smaller correlation with the CC-L ground-truth. We therefore used the complete breadth of the swath to generate CCClim despite the quality degradation towards the edges of the swath.

The final product of the framework published in Kaps et al. (2023a) is the RF model, which can predict relative cloud-type amounts using low-resolution cloud properties as inputs. For details on RF training and performance evaluation see Kaps et al. (2023a) or the respective code via zenodo[1].

CCClim is created by using the exact same RF model used for the results in Kaps et al. (2023a) to predict cloud-type RFOs for the ESACCI dataset. Cloud type predictions are made for grid cells of $10 \times 10$ pixels, i.e. $0.5° \times 0.5°$ and the predicted cloud types are averaged to daily values on a $1°$-grid. Note that the RF model is not applied to the same grid resolution it was trained on. This is possible since the method depends little on horizontal resolution in this range, as shown in the supplementary material of Kaps et al. (2023a).

The RFOs are normalized by the sum of the eight cloud types, i.e. they are independent of the total cloud amount. The "undetermined" class consists largely of clear-sky but also pixels for which one or more retrievals failed for other reasons. This class is therefore not a suitable indicator of the total cloud amount. Furthermore, cloud-free grid cells are explicitly excluded from our analysis, as the RF could not process them.

The following samples were excluded from CCClim because of faulty retrievals:

- All of July 2010 because of faulty surface temperature retrievals

- Two days at the turn of the year 1985/1986

- Retrievals at the end of 1994 were faulty due to the orbital drift of the NOAA-11 satellite, leading to the removal of the last 115 days of 1994 (November and December are already not included in ESACCI)

- Extreme outliers and fully cloud-free cells, detected by *tsurf* $< 10\,\mathrm{K}$ or *ptop* $< 10\,\mathrm{hPa}$ (amounts to $\sim 11\%$ of data, $\sim 727$ Mio. grid cells)

## 2.3 Concept Reasoning

The framework used to produce CCClim has been developed primarily to facilitate climate model evaluation: it is meant to produce meaningful and self-consistent cloud-type amounts on horizontal and temporal scales similar to the output of current global climate models. The consistency of the derived cloud-type RFOs was validated against their related physical variables and those of the classes in the CC-L dataset. This showed that the cloud types in CCClim are consistent with the classes obtained with the CloudSat algorithm (Kaps et al., 2023a).

---

[1]https://zenodo.org/record/7248773

| Abbreviation | Cloud Type |
|:---:|:---:|
| Ci | Cirrus/Cirrostratus |
| As | Altostratus |
| Ac | Altocumulus |
| St | Stratus |
| Sc | Stratocumulus |
| Cu | Cumulus |
| Ns | Nimbostratus |
| Dc | Deep convective |

**Table 1.** Cloud types used in CCClim, same as used in the CC-L dataset.

Since RFOs are obtained for low-resolution grid cells instead of using a classifier for high-resolution image pixels, details are inadvertently smoothed out. Conversely, this approach compensates for having multiple cloud types in a single pixel, e.g. via overlapping, a cause of ambiguity in a 2D analysis. Neighboring pixels can reinforce information on the dominating overlapping cloud type or provide complementary single-layer information. The coarser approach, therefore, leads to predictions that are more relevant for large-scale systems, while implicitly retaining information from the underlying high-resolution 3D observations.

Training on MODIS data and application to ESACCI data might at first glance seem like an unnecessary error source. However, training the classifier on ESACCI data is far more difficult, as the satellites providing the AVHRR data used in ESACCI are not part of the A-Train constellation (L'Ecuyer and Jiang, 2010). In contrast to CC-L and MODIS (onboard Aqua), easy co-location of the CloudSat/CALIPSO and AVHRR data is therefore not possible. Furthermore, MODIS measurements are only available for the shorter measurement period ($\sim$ 20 years), which is significantly less than the 35 years covered by AVHRR data. Furthermore, omitting the second ML step and using the pixel-level results directly would increase storage requirements, not enhance comparability with climate model output and also be subject to the previously mentioned issues with overlapping or ambiguous clouds.

## 3 Structure and Features of CCClim

Each year of the dataset is saved to a separate `netcdf` file. Each CCClim sample contains the fractional amounts of the nine classes and the eight input features (Table 2). The samples are identified by date and location (latitude-longitude) on a 1° regular grid as averages of the ascending and descending nodes, resulting in a single mean value per day. The total amounts of each cloud type in CCClim are shown in Fig. 2(a). Additionally, CCClim contains on average 22.6% of the "undetermined" class, i.e. at least 77.4% cloud, which is roughly 10% and 7% more than reported by ISCCP and CC-L, respectively (Young et al., 2018; L'Ecuyer et al., 2019). Some of this overestimation is due to the removal of cells that are (almost) totally comprised of clear-sky ($\sim$ 11% of the data). To better quantify the impact of this filtering we include the corresponding total cloud fraction

| Abbreviation | Variable | Unit | Classif. Input | Regr. Input |
|---|---|---|---|---|
| *cwp* | total cloud water path (ice + liquid) | $\frac{g}{m^2}$ | ✓ | ✓ |
| *lwp* | liquid water path | $\frac{g}{m^2}$ | | ✓ |
| *iwp* | ice water path | $\frac{g}{m^2}$ | | ✓ |
| *cph* | cloud top phase | categorical | ✓ | |
| *cer* | effective cloud particle radius | $\mu$m | ✓ | |
| *cerl* | eff. radius liquid cloud droplets | $\mu$m | | ✓ |
| *ceri* | eff. radius cloud ice particles | $\mu$m | | ✓ |
| *cod* | cloud optical depth | 1 | ✓ | ✓ |
| *ptop* | cloud top pressure | Pa | ✓ | ✓ |
| *htop* | cloud top height | m | ✓ | |
| *ttop* | cloud top temperature | K | ✓ | |
| *tsurf* | surface temperature | K | ✓ | ✓ |
| *ceff* | cloud effective emissivity | 1 | ✓ | |

**Table 2.** Retrieved physical input variables for the ML methods. The classification inputs correspond to the variables available from the MODIS Cloud Product. The regression inputs are computed from the ESACCI and MODIS data. The published CCClim dataset contains the corresponding ESACCI regression inputs for each cell.

(*clt*) from ESACCI in CCClim. Taking into account the filtering, the CCClim contains an average $clt \approx 68\%$ due to the low base cloud amount in ESACCI of $\sim 64\%$, resulting in a significant overestimation of cloud cover in CCClim compared to ESACCI (discussed in Section 5). The classes are subject to a large class imbalance with the most common class (Sc) occurring more than 27 times more often in the global mean than the least common class (Dc). This class imbalance is largely learned from the CC-L data (see Fig. 2(b)). The regional distribution is shown in Fig. 3. All cloud types show distinct and physically plausible

regions of high occurrence. Since the color scale was chosen to show that Sc is the most common type almost all over the globe, regions with an increased probability of forming a specific cloud type are not visible. We solve this by showing in Fig. 6 the type with the highest value of $\frac{RFO_{cell}}{RFO_{mean}}$ in each grid cell, i.e. the highest cloud type fraction relative to its respective global mean value. This highlights for example the regions of increased St occurrence or Dc, even though these are the least common types in CCClim.


     As another example for analysis on climatological timescales, we show the full time series for all eight cloud types averaged over the southern hemispheric oceans (defined as all ocean grid cells in the latitude belt from 0°S-90°S) is shown in Fig. 4. All eight cloud types show a distinct seasonal cycle. The spatial variability given by the shading in Fig. 4 does not change noticeably over the years. Figure 5 shows the average seasonal cycles calculated as the mean of each calendar day averaged

over the full 35-year period. Comparing the mean seasonal cycle to the seasonal cycle from individual years shows that the

relative deviation from this average cycle is typically smaller than 20%.

## 3.1 CCClim classes

While an exhaustive analysis of the classes and related processes is beyond the scope of this paper, we highlight some charac-
teristics of each cloud type in CCClim.

The Sc type is the dominating class in most regions, with a median global RFO of $0.31$. Relative to its total amount, Sc shows
little seasonal variability. Sc is subject to confusion which is the less common St type at all stages of the cloud classification:
the CC-L dataset used as ground truth has trouble distinguishing between St and Sc due to the small footprint of the active
sensors. The pixel-wise classifier and RF model propagate this uncertainty to CCClim, where St and Sc are predicted for simi-
lar conditions. Ci, the fourth most common class, appears most frequently in the tropics and subtropics, peaking in Southeast
Asia. This geographical distribution is in line with expectations from previous studies (e.g. Sassen et al., 2008). The As type
is prevalent in middle to high latitudes, especially over high-latitude land masses, with a high correlation with Ns. Although
the CCClim As distribution is similar to what is reported in literature (e.g. Sassen and Wang, 2011), results in polar latitudes
might be unreliable due to limitations of passive sensor retrievals in these regions. The amount of As is strongly modulated
by the seasons in both hemispheres, peaking in winter. In contrast, the Ac amounts peak in summer, with a slightly larger
seasonal amplitude in the Northern Hemisphere. For cumulus (Cu), the amplitude of the seasonal variation is smaller than for
most other cloud types. Interestingly, the Cu amount increases over the 35 years covered by CCClim, with its mean RFO over
the SH ocean increasing by $\sim 0.02 \approx 20\%$. Dc is the least common cloud type in CCClim because CC-L distinguishes sharply
between deep convective and multilayered cloud systems (L'Ecuyer et al., 2019). Even though Dc is rarely the dominant cloud
type in a coarse grid cell, Fig. 6 shows that there are distinct regions in which deep convection can occur more frequently (rel-
ative to the global mean RFO of this type) such as the intertropical convergence zone (ITCZ) over Southeast Asia or tropical
landmasses. Dc is subject to significant seasonal variations with values in summer about three times larger than in winter.

## 3.2 Process-based approaches

As an example for analyzing the impact of certain cloud types on climate, Fig. 7 shows the joint distribution of the short- and
longwave cloud-radiative effect (CRE) for each cloud type over the ice-free ($SST > 275$ K) oceans. The cloud radiative effects
are calculated from the top of the atmosphere (TOA) radiative fluxes provided as part of the ESACCI dataset by calculating
the differences between the clear-sky estimate and the corresponding all-sky value for short and longwave fluxes, respectively.
For this analysis (Fig. 7), only pixels with a sea surface temperature $SST > 275$ K are taken into account to reduce spurious
effects introduced by sea ice. Also, only cells with a cloud type RFO larger than the 84% percentile are considered to minimize
"contamination" with other cloud types. Note that despite this pre-selection of pixels every sample contains multiple cloud
types and thus derived values cannot be interpreted as absolute for the "pure" cloud type. We would like to note this is the
case for every sample in CCClim, i.e. in an approach like this, where cells with a relatively high amount of a cloud type are

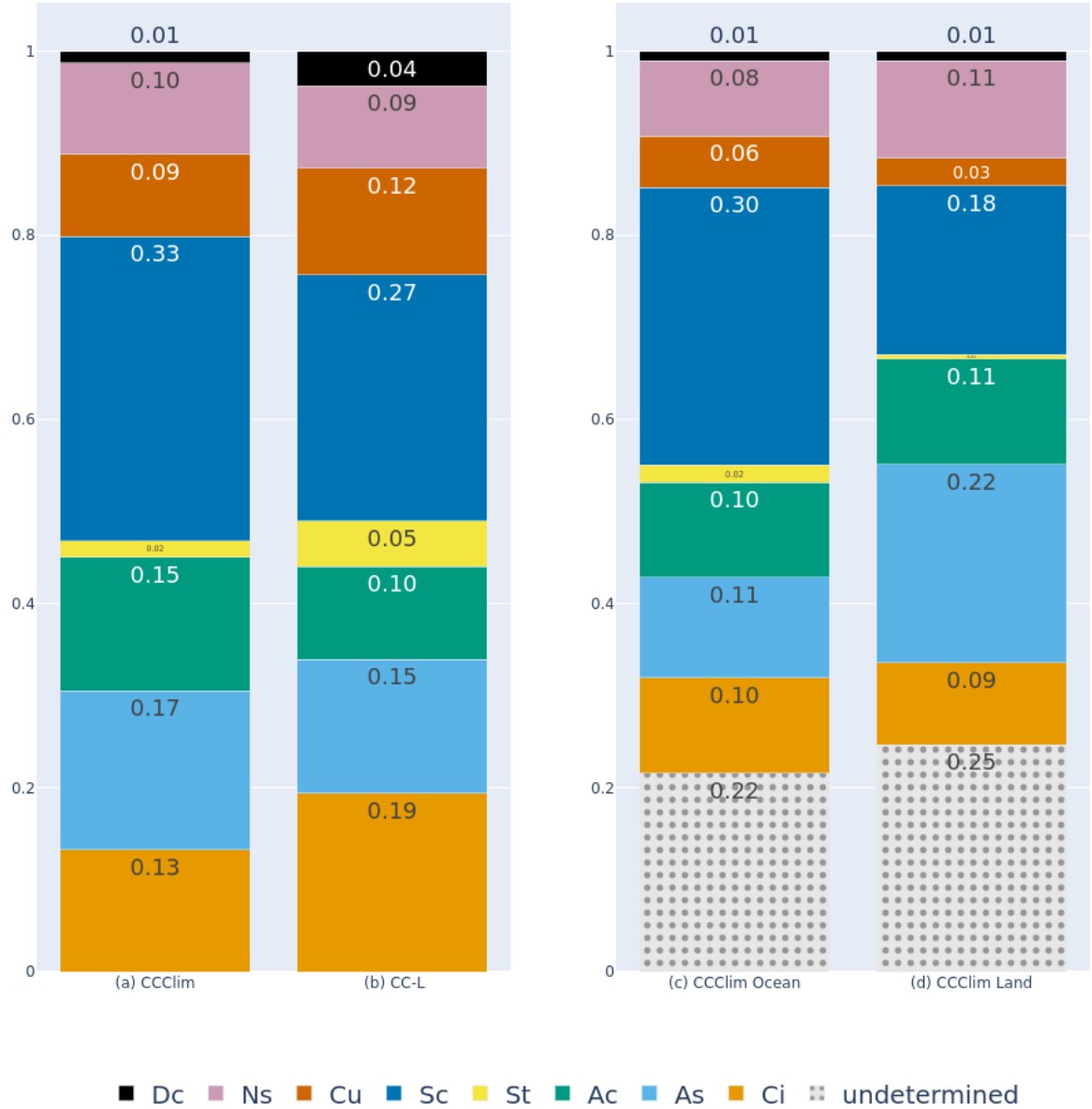

**Figure 2.** Relative occurrence of each cloud type in (a) CCClim and (b) CC-L for the year 2008 if clouds are present. CCClim results for (c) ocean-only and (d) land-only grid cells are shown in the lower row. The "undetermined" class includes clear-sky and is not included in (a) and (b) for comparability reasons, meaning that the appropriately weighted sum of (c) and (d) would give (a).

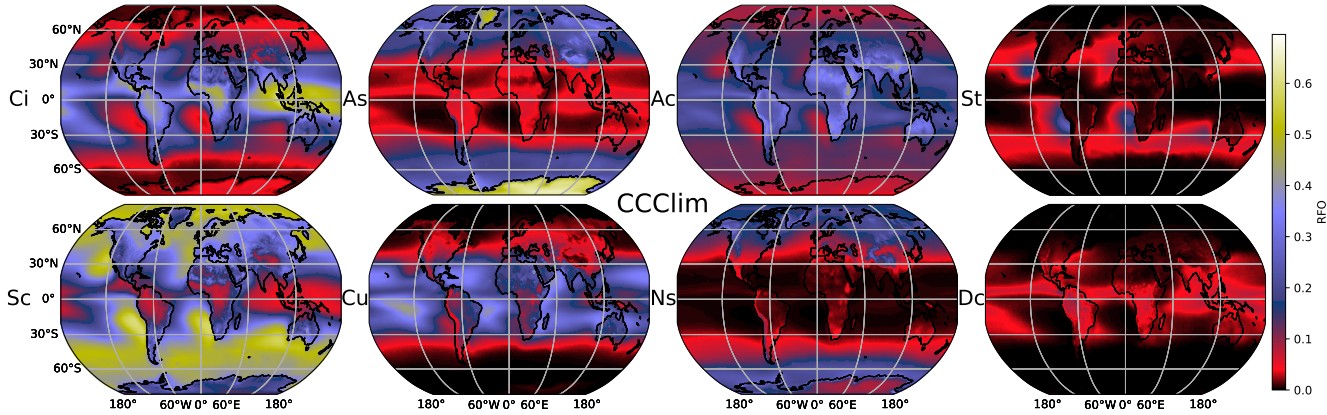

**Figure 3.** Average geographical distribution of the RFOs for all cloud types in CCClim.

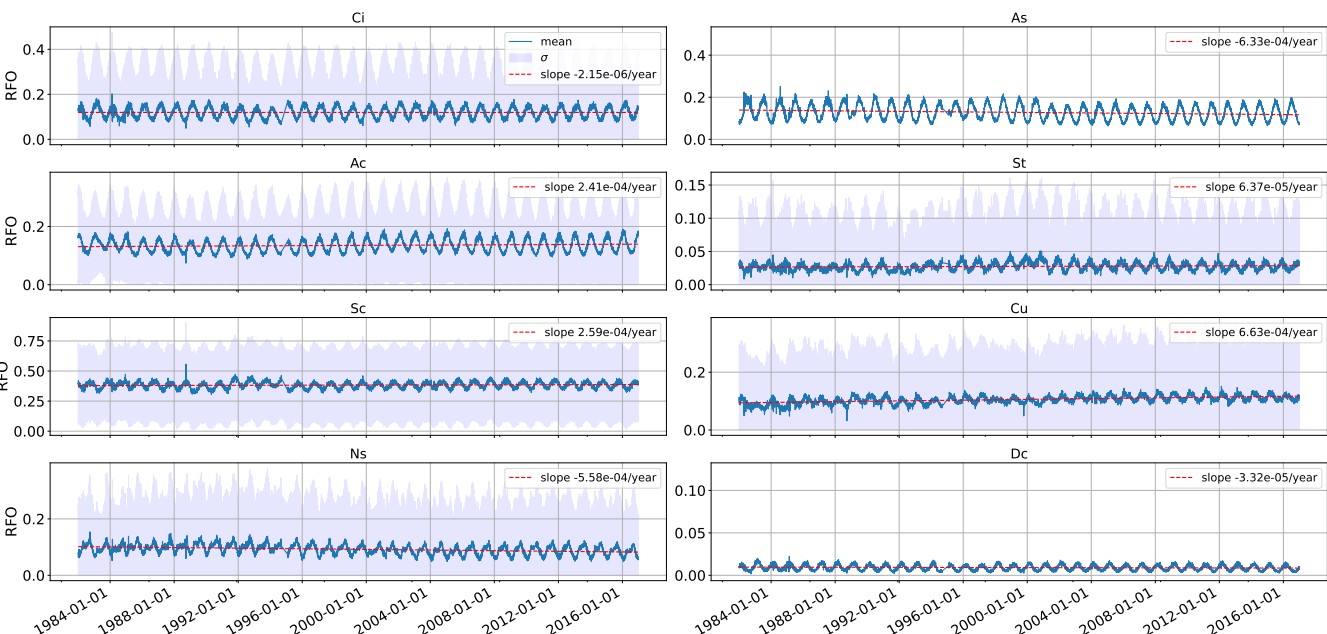

**Figure 4.** Time series of daily RFO values with the spatial standard deviation shown as shading for all cloud types averaged over the ocean in the southern hemisphere. All types show a consistent seasonal cycle and little anomalies and drift, as shown by the slope of the linear fit over the full period. Grid cells with a maximum RFO close to zero (1% quantile) are filtered out. Note that due to ESACCI being produced from measurements of multiple sensors, inconsistencies might be present in CCClim that make it unsuitable for reliable trend analysis.

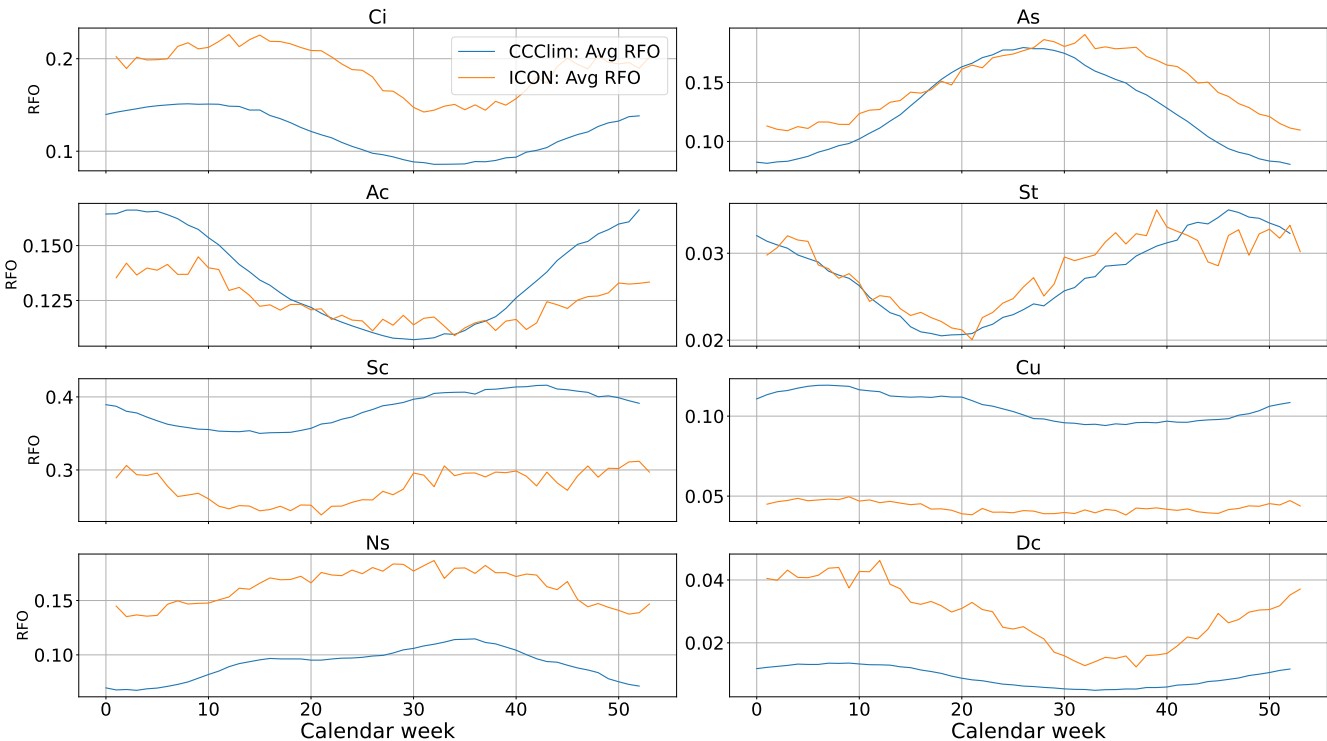

**Figure 5.** Example comparison of the mean cloud type RFOs by calendar week over the southern hemispheric oceans between CCClim (blue) averaged over the full 35-year period and an ICON-A simulation (orange, see Section 4) averaged over two years.

analyzed, other types can still significantly influence the values of the physical variables. For Fig. 7, using a percentile instead of an absolute threshold means that for types with globally low occurrence (St, Cu, Dc), contamination will be more of a factor. To illustrate this we will use the example of the Cu type, which displays one of the lowest longwave CRE in Fig. 7. Cu is strongly correlated with the "undetermined" class (not shown), which contains many clear-sky cases, partly explaining the small CRE values. Consequently, the CRE shown in Fig. 7 is not that of a cell filled only with Cu, but rather of cloud regimes associated with high amounts of Cu. Since these regimes tend also to contain clear-sky or low-top clouds other than Cu, the resulting CRE shown here is small. Another low-top cloud type, Sc, displays a larger CRE both in the short- and longwave spectrum. Again, these are influenced by co-occurring clouds resulting in the CRE of Sc varying with latitude. While Sc and St co-occur in the tropics and subtropics have a small longwave CRE, the correlation of Sc with Ac leads to a stronger longwave CRE in higher latitudes in the Sc-dominated cases. The stronger shortwave CRE of Sc in comparison to Cu can be attributed to Sc decks being more horizontally dense than the more cellular nature of shallow Cu. These relationships between the cloud types and their physical properties, as displayed with CCClim, might at first glance seem counter-intuitive. However, the interdependence of the cloud-type RFOs and their physical properties can also be beneficial. CCClim provides the context required to not only disambiguate the otherwise counter-intuitive relationships but also to highlight important cloud interactions. In Section 4 we

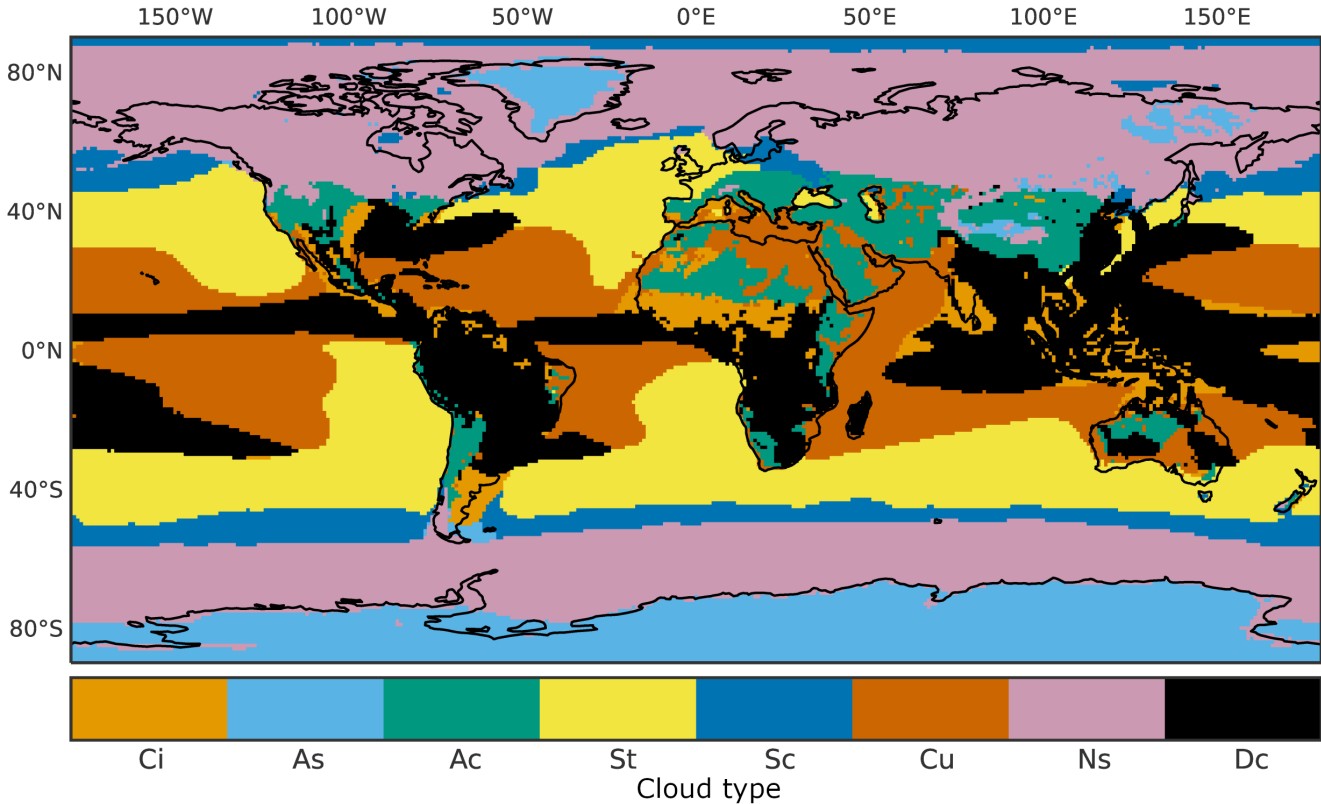

**Figure 6.** Cloud type with the highest RFO relative to its global mean value. This shows which regions have the most favorable conditions for the occurrence of a cloud type, independent of its total amount.

exploit this interdependence to better characterize the cloud types (see Fig. 10).

Mid-level clouds (Ac, As, Ns) display a more complex relationship between long- and shortwave CREs. Ac mainly differs from Cu and St through a larger longwave CRE, which is due to the higher and thus colder cloud tops. Ns and As, however, are distributed along an almost constant longwave CRE of approximately $32 \pm 5\,\mathrm{Wm}^{-2}$, with a peak at very small shortwave CRE values. An analysis of the associated cloud properties reveals that the clouds responsible for the weak shortwave CRE are located particularly at higher latitudes where As, Ns and Sc dominate cloud cover. Here, CCClim typically displays *iwp>lwp*, with small *cod*. This suggests optically relatively thin mixed-phase or ice clouds, which is qualitatively consistent with the observed CRE. The strong longwave CREs of the high-top clouds Dc and Ci are in line with expectations and very similar due to the high geographical correlation of the two types.

Another example for a process-based analysis of the CCClim data employs additionally $SST$ and $\omega_{500}$ from co-located ERA5 reanalysis data (Hersbach et al., 2020). $\omega_{500}$ is used as a proxy for the dynamical regime (large-scale circulation) and $SST$ as a proxy for the local thermodynamic conditions (Bony et al., 2004). The results are shown in Fig. 8. The maximum

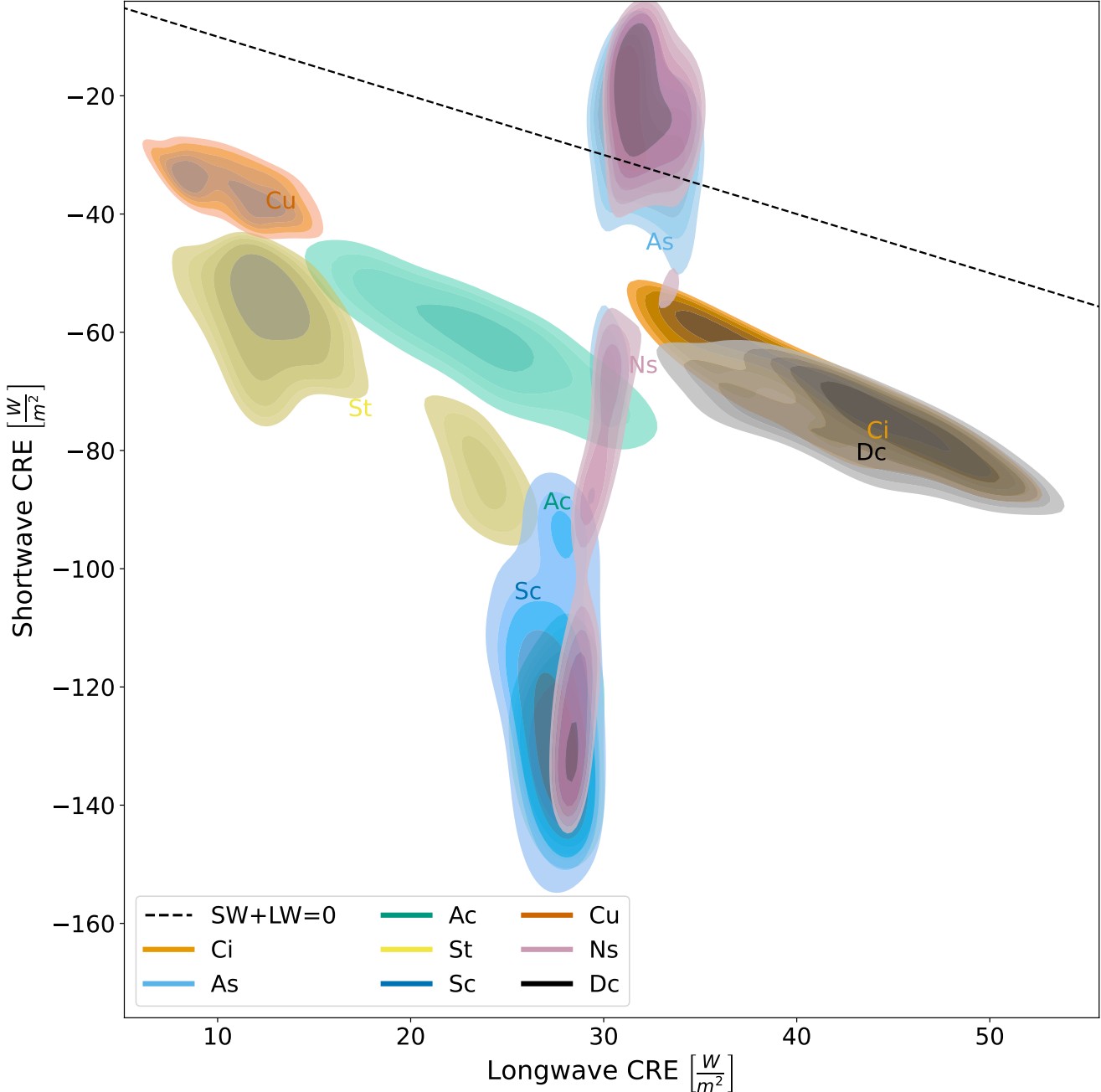

**Figure 7.** Longwave and shortwave cloud radiative effect (CRE) of the cloud types averaged by calendar month as kernel density estimates. The outer-most density level contains 30% of the probability mass of the samples per cloud type. Since each sample contains fractional amounts of multiple cloud types, the CREs are influenced by other clouds in the same cell. This influence is stronger if the type appears less often overall. We only include pixels over ocean with a sea surface temperature above 275 K to reduce possible spurious effects of sea ice. The cloud-type abbreviations are placed at the respective median CRE values. Due to filtering and spatial/temporal averaging, roughly $333,000$ samples are available, of which for each cloud type the $5 \cdot 10^4$ largest are sampled, corresponding roughly to the $84\%$ percentile.

occurrence of a cloud type in this phase space quantifies the conditions favorable for the formation of this type: while it is known that Sc occurs frequently in the large-scale subsidence regions over the subtropical oceans, we can narrow down the dynamical regime of maximum Sc occurrence to $SST^{Sc} \approx 299$ K and $\omega_{500}^{Sc} \approx 0.008 \, \frac{\mathrm{Pa}}{\mathrm{s}}$. The mid-latitude regions ($30°$ to $60°$) also show a significant amount of Sc for ascending motions over cold ocean surfaces ($SST^{Sc} \approx 275.4$ K, $\omega_{500}^{Sc} \approx -0.024 \, \frac{\mathrm{Pa}}{\mathrm{s}}$).
This Sc distribution is consistent with ISCCP data (Young et al., 2018), but the dynamical regime is not typical for Sc, such that mixed cloud regimes (Sc + As/Ns) can be inferred for this region. In contrast, Dc clouds are prevalent particularly in tropical regions with $SST^{Dc} \approx 302$ K and display large ascending motions with $\omega_{500}^{Dc} \approx -0.056 \, \frac{\mathrm{Pa}}{\mathrm{s}}$. Cirrus clouds often appear adjacent to deep convection in this phase space, presumably representing anvils or remnants of strong convection, as they show similarly warm sea surface temperatures ($SST^{Ci} \approx 301$ K) but rather descending air associated with the outflow regions of deep convective cells ($\omega_{500}^{Ci} \approx 0.004 \, \frac{\mathrm{Pa}}{\mathrm{s}}$). Ac clouds show a distribution similar to Dc and Ci in the $SST^{Ac}$ - $\omega_{500}^{Ac}$ space, corresponding to the Ac known to develop in the tropics from air detraining from deep convective systems.

## 4   Evaluation of global climate models

Comparison with CCClim can be a new avenue to evaluate clouds in climate models. To compare climate model data to CCClim, cloud-type distributions can be obtained from the climate model output using the same trained RF (Fig. 1) used to create CCClim from ESACCI. As an example, we applied the RF to output from the atmosphere component of the icosahedral nonhydrostatic model (ICON-A, version 2.6.1, untuned) (Giorgetta et al., 2018). Since not all required variables are available in the ICON-A standard output, we obtained effective particle radii and optical thickness at $2.32 \, \mu\mathrm{m}$ wavelength from the radiation parametrization (Pincus and Stevens, 2013). This waveband corresponds closely to the absorbing band used for the MODIS retrievals of *cod* and *cer*. Note that the optical depth computed here is directly related to the column-integrated water and does not depend on any overlap assumptions. If this is not the case for a different model an additional correction might have to be applied. The cloud top was determined as the level where the cumulative cloud optical depth of the layers above exceeds $cod = 0.2$, to be close to the detectable threshold by passive sensors (Eliasson et al., 2019). We used 3-hourly instantaneous values regridded to a regular $1°$ grid from the native R2B5 icosahedral grid (80 km) as input to the RF. Figure 9 shows the geographical distribution of the relative fraction of each cloud type averaged over a 2-year ICON-A simulation compared with CCClim. The cloud type distributions in both datasets are largely in agreement, but it is evident that low-top clouds (Sc, Cu, St) are underestimated in the simulation, in favor of Ns and Dc. This is consistent with the known underestimation of low-level marine clouds in ICON-A and many other global climate models (Crueger et al., 2018). The deviations will partly be due to erroneous cloud representation in ICON and partly due to inherent differences between model and satellite data. The contributions of either have yet to be determined. While an instrument simulator could be used to reduce systematic differences between ICON-A and MODIS for some of the variables, such software is not available for ICON-A. Also, because we are using an untuned version of ICON-A, no conclusions should be drawn from our analysis regarding the general performance. The following analysis is therefore intended purely to demonstrate how CCClim can be used for climate model evaluation, once all required variables are provided as instantaneous output by a climate model.

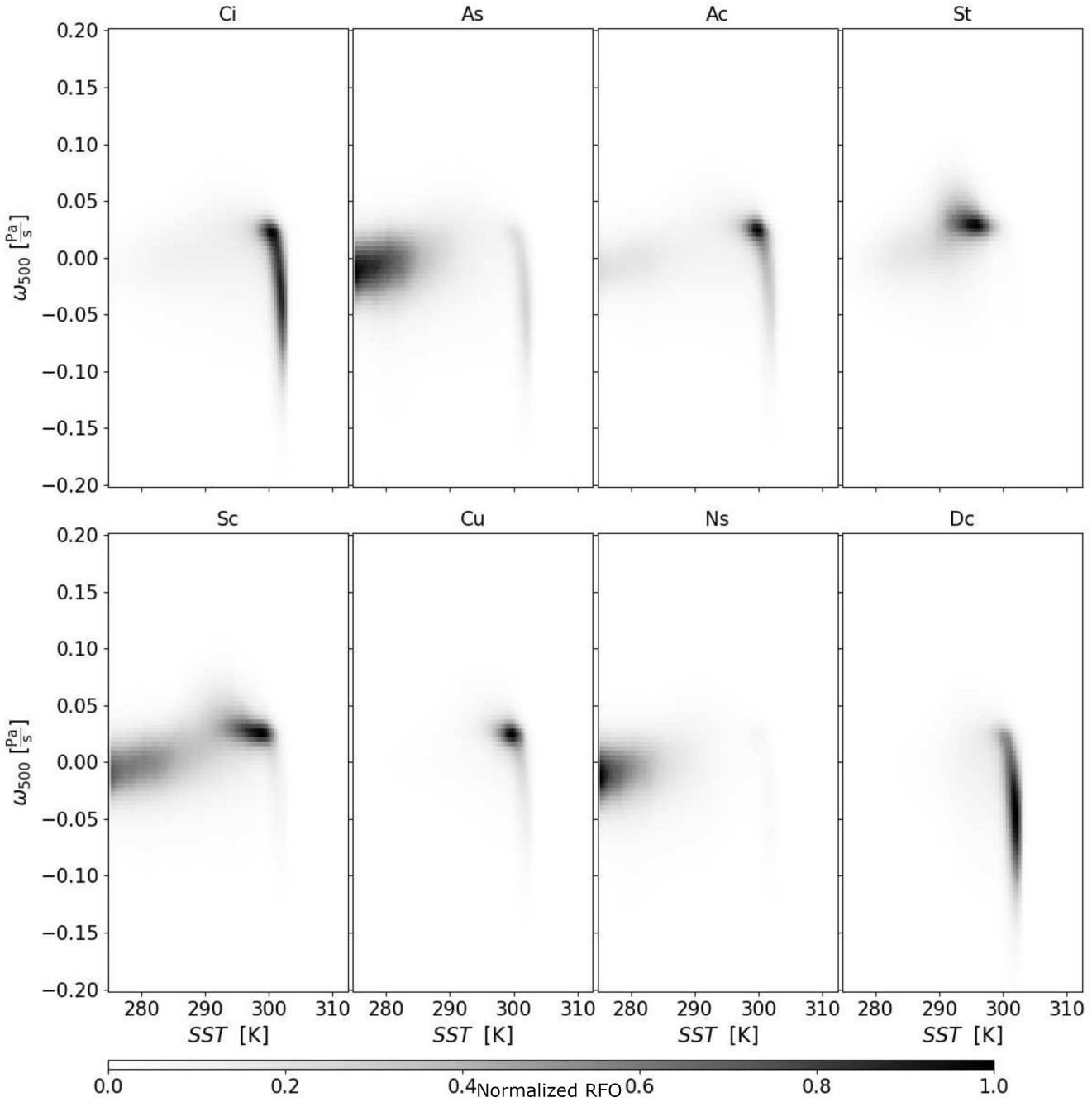

**Figure 8.** Distribution of each CCClim cloud type in $SST$ - $\omega_{500}$ space. For each grid point and type the RFOs are summed up and scaled by the maximum of this sum. The resulting distribution is therefore independent of the global amount of the respective type. These distributions can be used to infer the dynamical regimes favorable for the occurrence of each cloud type. As in Fig. 7, samples are averaged by calendar month with cells with $SST < 275$ K excluded. The largest $5 \cdot 10^7$ samples are used in terms of each cloud type RFO.

A high cirrus fraction is usually associated with a strong long-wave warming effect. However, both aspects of the CRE have been reported to be smaller in ICON than in observations (Crueger et al., 2018; Gettelman et al., 2020). We also observe smaller CREs in the output of this specific simulation. With respect to CCClim the simulation also exhibits an increase in the RFO of higher clouds with significant ice content (Ci, Ns) and a decrease in Sc and Cu. Figure 9 does show an increase in deep convection and we noted an almost binary distribution of either a very high or very low Dc fraction per cell (not shown). This could indicate that convection in the simulation only develops under specific conditions, but is strong and stable if it does get started. The fact that we have a higher average cloud top means that non-convective processes play a significant role in forming high clouds here. The resulting Ci clouds are thin as most cells that contained $> 90\%$ Ci have an ice water path $iwp < 10 \frac{g}{m^2}$ (Fig. 10), corresponding to a cirrus-attributable optical depth of $cod < 0.1$ (Heymsfield et al., 2003). Similarly, an overestimation of high, thin clouds has been found in other climate models (Kodama et al., 2012). The Ci-property analysis resolves the apparent contradiction with the decreased longwave CRE, as subvisible Ci have a negligible radiative impact (Spreitzer et al., 2017; Turbeville et al., 2022). As the example ICON-A run used here has not been tuned, going into further details regarding the processes involved in the formation of large areas of thin Ci are not sensible.

Conditioning cloud property distributions on cloud types like in Fig. 10 can also facilitate the analysis of distinct cloud processes as similarly demonstrated for ISCCP weather states (Tselioudis et al., 2013; Oreopoulos et al., 2014, 2016). Using the conditional distributions, we find that in ICON-A both, Ci and Sc, tend to be much thinner than in CCClim, indicated by the ice/liquid water path values, respectively. Furthermore, even in cells containing mostly Sc, Ac is often present. This is not evident from the regional distributions (Fig. 3), which show that the Ac amount decreases over the oceans. We can deduce that this mixture of Ac and Sc occurs at the interface of more convective regions near the equator and regions of large-scale subsidence in the subtropics.

To apply these comparisons to climate models more generally, the output of all eight quantities as instantaneous values would be required, preferably using instrument simulators providing output similar to ESACCI to account for, for example, temporal and spatial sampling of the satellite. For a comprehensive evaluation of a climate model, corresponding periods or climatological averages of 20 years or more should be compared.

## 5   Capabilities and Limitations of CCClim

The CCClim dataset presented in this paper allows for the investigation of clouds grouped by WMO-like cloud type with a long temporal coverage and high spatial resolution as daily samples. Using multiple cloud properties to define the classes makes them physically consistent and well-aligned with the morphological WMO genera. We showed that categorizing complex atmospheric data into established types is more expressive and interpretable than individual cloud properties. This is shown exemplary by analyzing the properties of different cloud types for given atmospheric conditions, increasing insight into important processes driving cloud development. As an example of how to evaluate global climate models with CCClim, we showed a comparison with cloud types obtained using the same method from output of the ICON-A model.

Our results show that the cloud types in CCClim have consistent seasonal variations, sensible regional distributions and little

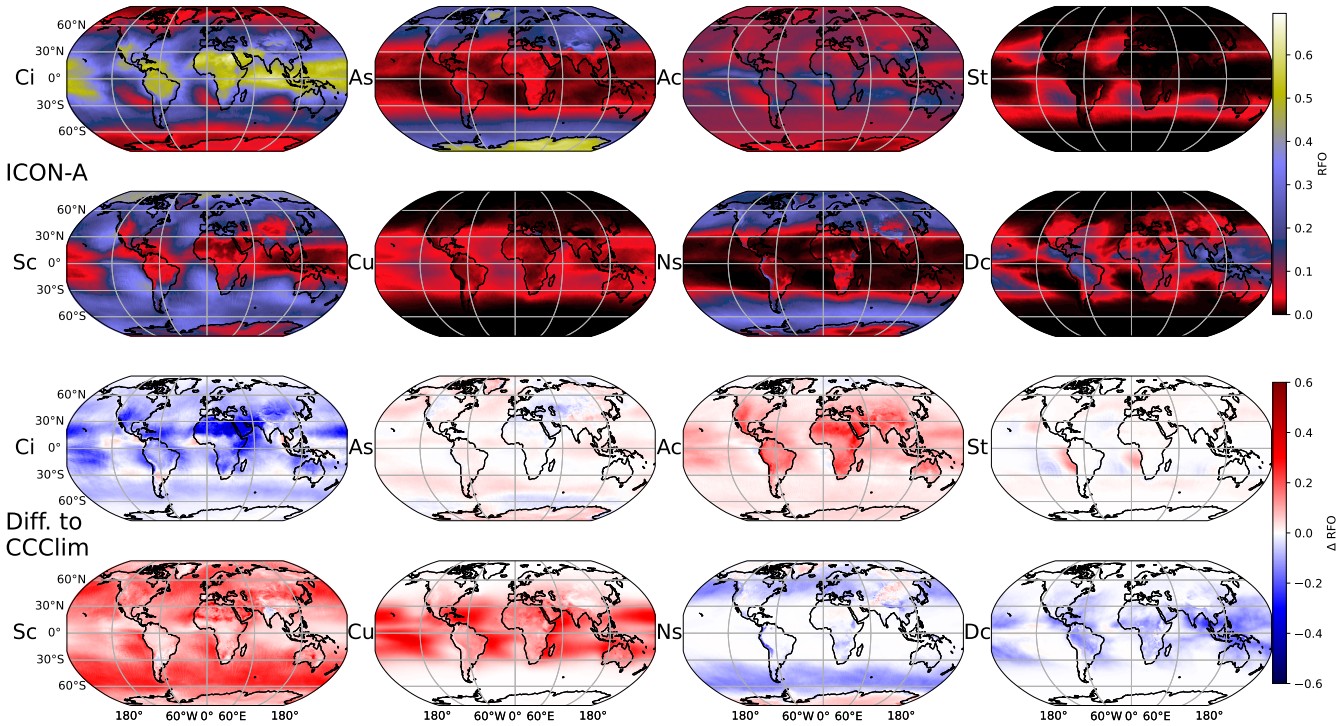

**Figure 9.** Average cloud type distributions obtained with our RF model from two years of ICON-A output (top rows) and the differences to the CCClim distributions (bottom rows), where positive (red) values denote a higher value in CCClim and negative (blue) a higher fraction in ICON-A. Note that these values show the fractions of existing clouds, irrespective of cloud cover, thus the high fraction of Ci over northern Africa in ICON, which cause the extreme deviations at the end of the color scale.

drift over the complete period. This makes CCClim suitable for statistical analyses of clouds and enables quantification of seasonal cycles of WMO-like cloud types on a global scale. As the cloud types have been learned from the CC-L dataset, its errors have been propagated. This applies to the distinction between St and Sc, such that the CNN classified many St clouds as Sc, leading to a small fraction of St in CCClim. This problem could partly be explained by difficulties of CloudSat/CALIPSO

in measuring low-level clouds in overlapping conditions, where the lidar is attenuated by higher clouds and the radar is affected by ground clutter (Marchand et al., 2008). For some applications, it might therefore be better to combine St and Sc into one new class. Also, since CC-L labels are only available for nadir-near MODIS pixels, uncertainties might be introduced by labeling off-nadir pixels in the classification stage. A correction, however, is non-trivial (e.g. Painemal et al., 2021) and no concrete solutions are available in literature. Simply discarding pixels that are "far" away from the center reduces the data coverage and

results in a significant reduction in performance. Even though the viewing angle effect is very hard to quantify, it is expected to be at least somewhat mitigated through the spatial and temporal aggregation in CCClim (e.g. Maddux et al., 2010; Bennartz and Rausch, 2017). Furthermore, since CCClim contains a very low absolute amount of the Dc type, we recommend focusing

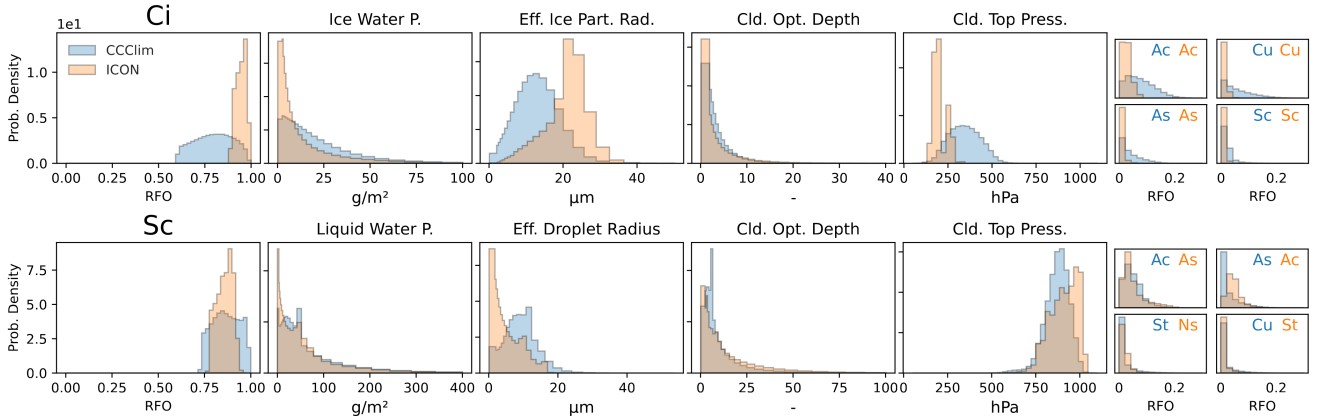

**Figure 10.** Analysis of cells "characteristic" for Ci/Sc in ICON and CCClim in terms of probability densities. Characteristic cells are defined to have at least $85\%$ of the chosen cloud type (Ci or Sc) and/or the "undetermined" class and to also have a higher cloud RFO than the respective global median RFO. The resulting distribution of the class is shown in the leftmost panel. The four middle panels show cloud-relevant properties in the characteristic cells. The rightmost panel shows the distribution of the four cloud types that coincide most often with Ci/Sc in the respective dataset. This allows characterization of the physical properties of the cloud types while being able to take into account the contributions of other classes in the cells.

on relative changes when studying this cloud type, like in Fig. 6. With this approach, the prevalence of deep convective clouds in, for example, the ITCZ becomes more apparent.

Comparison with other cloud-type statistics has shown that our method likely underestimates the amount of cirrus clouds. This underestimation already occurs in the pixel-wise classification step, even though plenty of samples with the Ci label are available in the training data. The underestimation is therefore not caused by the difficulty of passive sensors to detect thin cirrus clouds, but rather by how the retrievals deal with multi-layer clouds. Wang et al. (2016) found, that when a cirrus cloud overlaps with a low-level cloud, the resulting MODIS retrievals indicate a high cloud of medium optical depth. If the corresponding

CC-L label is Ci, the classifier learns that high-top, medium-thickness clouds are cirrus, leading to a combined confusion with As and Ac of 21%, i.e. 21% of Ci are falsely labeled as As or Ac. A similar phenomenon was found when the properties of ISCCP regimes were compared with active sensor measurements, indicating that this is a common problem when using passive sensor data for cloud classification (Haynes et al., 2011). Due to the added difficulty of detecting thin clouds over ice surfaces, it is advisable to exclude polar regions from more detailed analyses using global data, similar to what has been done for the

examples shown here (Figs. 7, 8).

    Grid cells with many erroneous cloud property retrievals, which are often a result of clear-sky, have been removed as the RF can not process them. This reduces the amount of clear-sky contained in CCClim (implicit via the "undetermined" class). These cells can also not simply be labeled as clear-sky as we do not know why the retrievals failed. Furthermore, the sum of the cloud-type RFOs in CCClim is higher than the cloud fraction obtained from ESACCI. In combination with the high-latitude

clouds with vanishing short-wave CRE, this suggests a bias induced by applying the RF on a different domain than it was

trained on. We suspect that some of the As clouds in high latitudes might be clear-sky, but differences between the retrievals of MODIS and AVHRR lead to different results. Therefore, the ESACCI cloud fraction (*clt*) is provided in CCClim to enable analysis of the cloud types in the appropriate cloud cover context. The RF used to predict the cloud-type distributions was trained using only daytime observations. However, CCClim was created using all viable samples from ESACCI, including

nighttime observations. While the physical relationships between cloud properties and types are expected to be the same at night, the ESACCI nighttime retrievals are considered experimental (Stengel et al., 2020). Therefore, potential errors in the nighttime retrievals are propagated to CCClim. In most cases, sampling biases such as the incomplete sampling of the diurnal cycle are expected to be outweighed by daily and seasonal variability. Note, also, that CCClim is not suitable for detecting trends in the cloud-type distribution in response to climate change effects, as the underlying ESACCI data are deliberately

adjusted to produce a long-term stable dataset (Sus et al., 2018).

     The few datasets attempting to assign objective WMO-like cloud types often disagree on important details. For example, while Stubenrauch et al. (2006) are in good qualitative agreement with the ISCCP cloud distributions, they find, for instance, a much higher cirrus fraction in the tropics. For comparison, CCClim is less affected by problems common to these purely passive-sensor-based datasets, such as dealing with multilayer clouds (Li et al., 2015). Cluster-based datasets are however not

suitable for a direct comparison with CCClim, as even intercomparison of datasets produced with unsupervised methods is difficult since the derived clusters do not have a common physical basis across all datasets. The comparisons made between such clusters and classical ISCCP regimes show disagreements that can be attributed to differences between (active and passive) sensors, definitions of cloud types and distinguishing between individual cloud layers (Li et al., 2015; Marchant et al., 2020; Kurihana et al., 2022). Therefore, a product such as CCClim covering a long time period can take advantage of synergy effects

to address some of the limitations inherent to passive and active satellite sensors and provide a more consistent dataset than available from individual instruments. Even though subjective thresholds are being used in the fuzzy logic classifier of CC-L, combining the different cloud observations from space radar, lidar and from passive instruments is a clear advantage over classification methods that do not use active sensor data such as such as many unsupervised approaches. Using additionally other kinds of passive measurements could further improve the results. Using ML for this combination of sensors avoids introducing

any further subjective biases. The resulting dataset is easy to use in terms of physical interpretation of the cloud types, data volume, horizontal grid and common metadata.

*Code and data availability.* CCClim can be downloaded from zenodo without restrictions under the DOI 10.5281/zenodo.8369201(Kaps et al., 2023b). The code for training the RF and its validation is also on zenodo (10.5281/zenodo.7248772) or our GitHub at https://github.

com/EyringMLClimateGroup/kaps22tgrs_ml_cloud_eval/. The code for the creation of CCClim is accessible on zenodo under the DOI 10.5281/zenodo.10279991. CUMULO is available from https://github.com/FrontierDevelopmentLab/CUMULO.

The ESACCI dataset and related information are available under the DOI 10.5676/DWD/ESA_CLOUD_CCI/AVHRR-PM/V003 (Stengel

et al., 2019). The ERA5 data was downloaded from the Copernicus Climate Change Service (C3S) (2023) and is available under the DOI 10.24381/cds.bd0915c6 (Muñoz Sabater, 2019).

*Author contributions.* AK, AL and VE developed the concept. AK trained the machine learning models and carried out investigation of data and results. RK performed the ICON simulation. MS provided scientific advice for the ESACCI dataset. AK and AL prepared the manuscript with contributions from all authors.

*Competing interests.* The authors declare that they have no conflict of interest.

*Acknowledgements.* AK and VE were funded by the European Research Council (ERC) Synergy Grant "Understanding and modeling the
Earth System with Machine Learning (USMILE)" under the Horizon 2020 research and innovation programme (Grant Agreement No. 855187). AL was funded by the ESA Climate Change Initiative Climate Model User Group (ESA CCI CMUG). MS was supported by the European Space Agency (ESA) through the Cloud_cci project (contract no.: 4000128637/20/INB). This work used resources of both, the Deutsches Klimarechenzentrum (DKRZ) granted by its Scientific Steering Committee (WLA) under project ID bd1179 and the supercomputer JUWELS at the Juelich Supercomputing Centre (JSC) under the Earth System Modelling Project (ESM). The results contain modified
Copernicus Climate Change Service information 2020. Neither the European Commission nor ECMWF is responsible for any use that may be made of the Copernicus information or data it contains. .

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
