# Peer review of "Characterizing clouds with the CCCLim dataset, a machine learning cloud class climatology"

_Earth System Science Data, 2023_

## Referee Comment (RC2)

**Review of the manuscript 'Characterizing clouds with the CCClim dataset, a machine learning cloud class climatology'**

Peter Kuma

Department of Meteorology (MISU), Stockholm University, Stockholm SE-106 91, Sweden

7 December 2023

Dear editor and authors,

The authors present a dataset of cloud types derived from satellite data using supervised machine learning previously described by Kaps et al. (2023a). This complements existing cloud type datasets such as ISCCP and those derived using unsupervised machine learning. I find the study well-presented and the discussion and conclusions meaningful. However, there are some factual errors, some details require further clarification, all limitations should be properly discussed, and potentially more validation would be beneficial. My comments are detailed below.

Kind regards,

Dr. Peter Kuma

**General comments**

Calling the CC-L cloud types WMO cloud types is misleading. The WMO cloud genera are defined using ground-based visual observations, as standardised by the WMO International Cloud Atlas. While CC-L definitions of cloud types aim to be similar to the WMO cloud genera, they are not the same. The deep convective (Dc) type is also not a WMO cloud genus. I think that the authors should be more clear about this.

There is a considerable overlap between the presented work and Kaps et al. (2023a). It is not entirely clear if the authors use the exact same methods as in the previous paper, or if anything is different. I think that the authors do not necessarily have to duplicate the description of the methods in this manuscript, and can instead reference the previous paper, which is really a pre-requisite for understanding this manuscript.

L177–178, Fig. 4 caption: The Cloud_cci AVHRR L3U PM daily data are a composite of instantaneous values and not a daily mean, because the satellite overpass time is always at about the same local time, and the data have not been corrected with a diurnal cycle model. Stengel et al. (2020): 'All data are collected on two processing levels: (a) Level-3U, which represents daily composites of non-averaged data collected on a global latitude–longitude grid with 0.05° resolution and'. Unless some more sophisticated processing is done on the input data, the CCClim dataset also represents daily composites of instantaneous values.

It is not clear whether the results only represent daytime clouds or nighttime as well. The authors say that the Cumulo dataset is only available for daytime, but they do not say whether they apply the RF on nighttime grid cells from Cloud_cci and ICON-A. If they only apply it on daytime, it should be made more clear that the results can be biased toward daytime clouds and not representative of polar night conditions. If they apply it on nighttime too, this should be mentioned as a limitation due to the fact that this is assuming that the statistical links between the input physical quantities and cloud types applying during daytime also apply during nighttime (reference training data for nighttime are lacking in this case).

Because the comparison between cloud types derived from ICON-A and Cloud_cci is between 3-hourly instantaneous output of the model and daily composites of AVHRR measurements, it is not necessarily comparing the same time of day at local time. Because clouds tend to exhibit a diurnal cycle, this can introduce biases. This should be at least mentioned as a limitation. Also comparing 35 years of CCClim from AVHRR with 2 years from ICON-A is not necessarily a good idea because the global climate has changed during the last 35 years. Comparing the same 2 years of both would be better.

One potential limitation that the authors do not mention is that cloud-related quantities such as cloud top pressure and cloud optical depth are not always directly comparable with those derived from radiance measured by a satellite instrument. For this reason, simulators for ISCCP and MODIS such as COSP exist. Because the RF takes these quantities as a input from the observations and the model without using an instrument simulator, an artificial bias can be introduced in the comparison.

The authors should include short sections in the Methods describing the Cumulo dataset and ICON-A. The section on CC-L should at least briefly describe how the CloudSat–CALIPSO cloud types are defined in the original dataset.

A global regular angular (longitude—latitude) grid has unequal area of grid cells by latitude. This means that grid-cell averages are calculated over larger areas over polar latitudes than equatorial latitudes. This can have an effect on a predictor, such as the RF used on the coarse-grained Cloud_cci data or ICON-A data. I think that this should be at least discussed.

I think it would be possible to use a single machine learning model to determine the cloud types instead of two (IResNet on the fine scale followed by RF on the coarse scale). One could train a CNN to predict coarse grid cell cloud type fractions by training on grid cells where CC-L data are available (averaged over the coarse grid cell and represented as fractions of cloud types). It seems to me that the two step approach can be both more complex and less stable. Can the authors comment on this?

'CloudSat labels' in the text and figures should be really called CC-L labels or CloudSat–CALIPSO labels, since they are based on measurements from both satellites.

I suggest that the authors use terms 'cloud type' and 'cloud class' consistently throughout the manuscript. They seem to be used interchangeably, but it would be better to use either only one of them, or define the distinction between them.

The active satellites sensors used as a reference for the CC-L cloud types have known problems with detecting low clouds. The CloudSat radar is affected by ground clutter, and the CALIPSO lidar signal is often attenuated by overlapping clouds before reaching mid and low-level clouds. These problems in turn also affect the presented dataset, and should be mentioned as a limitation.

The authors train the RF on MODIS but apply it to Cloud_cci. It is not clear if this produces any side-effects. Ideally, the authors should compare the results of applying the RF on MODIS and Cloud_cci for the same testing time period (excluded from training).

Fig. 4: Long-term AVHRR datasets are usually not very well suited for determining trends due to discontinuities in the orbital parameters of the satellite series. This limitation can also translate to any derived dataset. Can the authors comment on whether the slopes shown in the figure are reliable? The authors mention this on L339, but it would be better to also caution the readers in the context of Fig. 4, and also on L8–10.

L14–15: **or its radiative effects**: This could be a problem with reanalyses which usually parametrise clouds, which can result in radiation fluxes not corresponding to the actual cloud cover.

L224: **ice-free oceans**: It would be better to say 'ice-free (SST > 275 K) oceans', since the authors are not actually

using sea ice information to exclude grid cells, but a temperature condition which also excludes some ice-free ocean grid cells.

L275: **This waveband corresponds closely to the absorbing band used for the MODIS retrievals of cod and cer.**: But what about AVHRR? Since the comparison here is between cloud types derived from ICON-A and from AVHRR.

L290–291: **With respect to CCClim the simulation also exhibits an increase in high cloud RFO (Ci, Ns)**: Ns is classified as a middle level cloud in the WMO International Cloud Atlas.

L351–352: **there is arguably no better way to obtain cloud observations from space than combining radar, lidar and passive instruments.**: I think that this is merely stating the obvious. Since there are currently no other satellite instruments relevant to cloud measurements other than lidar, radar and passive instruments, combining them necessarily has to give the best opportunity to obtain cloud measurements. One could also consider other kinds of passive measurements such as those by geosynchronous satellites, multi-angle measurements (MISR), passive microwave, Doppler lidar (Aeolus) and passive instruments on deep space satellites (DSCOVR). Including those in a combined analysis could produce more accurate results. The fuzzy logic classifier used in CC-L could have many different alternatives, for example using unsupervised learning or by supervised learning using ground-based observations. I do not think that one could say that it is the best, but rather one of many options of how to classify clouds.

**Technical comments**

Greater attention is needed to make the plots accessible to people with colour blindness. For example, in Fig. 1, 2, 6 and 7 some of the cloud type colours could not be distinguished. I recommend testing this for example with KMag on Linux.

L44: Missing space in 'CCClim(Kaps et al.,'.

L105: I think that the official name of the dataset is 2B-CLDCLASS-LIDAR.

L133: Parenthesis not closed.

L141: **grid boxes**: Probably better to say 'grid cells' for consistency with the rest of the manuscript.

L190: **Fig.6**: Missing space.

L240, 268, 344: **Tropics**: Usually not capitalised.

L277: **sensors(Eliasson et al.)**: Missing space and missing year.

L294: **as most cells contained >90% Ci have an ice water path** $iwp < 10\frac{g}{m^2}$: as most cells *which/that* contained...

L355: This section should contain links to all of the datasets used, including Cumulo, ERA5, MODIS, 2B-CLDCLASS-LIDAR, Cloud_cci and ICON-A.

L372: It should be made clear that the DOI is a DOI.

Table 1: **cloud type**: Should be capitalised to be consistent with Table 2.

Table 1, Fig. 6: Capitalisation of the first word in the caption.

Fig. 2: This is more of a style issue, but pie charts are usually not considered a good visualisation method because the proportions are hard to compare. Bar charts usually provide a much better experience.

Fig. 3, 8, 9: Ideally, all colour bars should have labels.

Fig. 4: The plots should have y-axis labels.

Fig. 5: Plots should have a y-axis label RFO.

Fig. 5: **see 4**: 'see Section 4'.

Fig. 10: The plots are probably too small. I recommend dedicating more space to the figure. The axes should have better labels, i.e. not just the cloud type name, but also that it is RFO on the x-axis, and probability density on the y-axis.

---

## Author Comment (AC1)

**Response to the Reviews**

January 13, 2024

Arndt Kaps Deutsches Zentrum für Luft- und Raumfahrt e.V. (DLR) Institut für Physik der Atmosphäre, Oberpfaffenhofen, Germany Email: arndt.kaps@dlr.de

Earth System Science Data Manuscript Number: essd-2023-424

Dear Editor:

Thank you for guiding our paper through peer-review. We have addressed all of the referees' comments as detailed in our point-by-point replies below. In particular, we made the following main changes to the manuscript:

- We have clarified the relationship between the cloud types used by us and the WMO cloud genera, now using the terminology of "WMO-like" types;
- We have stated more clearly that our comparison to ICON-A data is only intended as an example and that comprehensive climate model evaluation would require more steps, such as using an instrument simulator;
- We mention some limitations of CCClim more prominently, such as the incomplete sampling of the diurnal cycle;

Yours sincerely, Arndt Kaps on behalf of the author team

**Response to Referees**

**Earth System Science Data**

| Manuscript: | essd-2023-424                                                                        |
|-------------|--------------------------------------------------------------------------------------|
| Title:      | Characterizing clouds with the CCClim dataset, a machine learning cloud class clima- |
|             | tology                                                                               |
| Authors:    | Arndt Kaps, Axel Lauer, Rèmi Kazeroni, Martin Stengel, Veronika Eyring               |
| Date:       | January 13, 2024                                                                     |

We thank the two referees for their helpful comments. In this document, we answer each point raised by the referees. The original referees' comments are given in **blue**, our answers in **black**. All line numbers refer to the "track changes" version of the revised manuscript.

**Referee: 1**

This paper is a noteworthy addition to the growing literature of applying ML to Earth Science remote sensing, in this case cloud remote sensing. As the observational systems change and certain capabilities are interrupted or discontinue, we need innovative methods to cover data gaps. AI and ML have a lot to offer in that regard.

My recommendation is to accept the paper with minor revisions. I was tempted to recommend a major revision because I think that the active dataset is not used to its full potential, but it'd be unfair to demand from the authors a paper of a different direction. So this paper is judged on the merits of the fundamental choices they have made. Still, there are some major issues on how ML was implemented (see below).

What makes this paper in my opinion less than it could have been is the information content of the active observations they decided to use: so-called "cloud types". There are two issues: (1) The main appeal of active observations is that they can resolve (in many cases) cloud vertical structure, so to choose just a cloud type "flag" (the only or most dominant cloud type in an active observation "ray" or profile) seems like the least interesting choice;

We thank Referee 1 for providing helpful comments to improve the manuscript.

The authors agree that taking full advantage of the vertically resolved cloud column would be more interesting. However, such an approach comes with several technical challenges. The goal of the first stage of ML, as explained in [1] is to effectively extend the coverage of the spatially extremely sparse CC-L product. Any information on the 3D structure inferred from 2D MODIS observations would introduce more ambiguity than working with column-representative cloud types. In fact, such an approach is currently investigated by collaborating researchers but with inconclusive results so far.

(2) Even these "cloud types" from 2B-CLDCLASS-LIDAR (CC-L) are taken too literally by the authors: they're not the WMO cloud types the authors imply them to be, but just cloud labels that may have some association; but I suspect frequently they do not have the morphological characteristics that surface observers use to classify clouds. More on this topic below. On cloud types:

The WMO has 10 main cloud classifications https://cloudatlas.wmo.int/en/cloud-classification-summary. html. There's no Deep Convection (DC) as in CC-L, but rather Cumulonimbus (Cb). There is also cirrocumulus and cirrostratus, which do not exist in 2BCL (presumably all under Cirrus). But the biggest tell of the lack of correspondence between CC-L and WMO is the small occurrence frequency of stratus (noted by the authors). If one checks daytime stratus occurrence over the ocean from surface (ship) observers in the Warren Atlas https://atmos.uw.edu/CloudMap/WebO/index.html, there is plentiful stratus in the extratropical oceans. So, the same way the 9 ISCCP cloud types defined by arbitrary boundaries in the TAU-CTP joint histogram cannot be taken as equivalent to the corresponding WMO cloud types (cloud morphology as appears from surface), mentioned by the authors, the CC-L cloud types can also be assumed to mean the same thing as the WMO classification. Another tell: the mall SW CRE peak the authors find for Ns which are optically thick rain-producing clouds.

We agree with the referee, that speaking of WMO cloud types is not fully correct and thus confusing. We

therefore applied the following changes listed below in the revised manuscript and now speak of "WMO-like" classes. The differences in naming conventions between CC-L and WMO have been clarified and the observed differences to the Warren Atlas mentioned. As there are no major conceptual differences between the CC-L classification and WMO genera from [6], deviations are likely caused CC-L's sensor characteristics. The inconsistency with the Ns CRE noted by the referee is probably not caused by these definitions but rather by the ML algorithms confusing Ns and As, as well as the frequent co-occurrence of Ns and As. The following changes have been applied to the text:

- (Abstract L5): ...eight major cloud types, designed to be similar to those defined by the World Meteorological Organization (WMO) ...;
- (Introduction L45-46): The cloud types contained in CCClim are physically consistent with most of the major cloud genera defined by the World Meteorological Organisation [7]...;
- (Data and Methods L85-90): These are used in CUMULO to provide target labels as WMO-like cloud types from CloudSat's 2B-CLDCLASS-LIDAR dataset (hereafter CC-L) [6]. We call the cloud types "WMO-like" because they are defined to correspond to eight of the ten WMO genera, with the Dc (deep convective) type replacing the WMO's cumulonimbus and cirrocumulus/cirrostratus being contained in the Ci (cirrus) type. While these classes are defined to be consistent with the WMO definitions, misclassifications can occur, caused for example by the small footprint of the active sensors;
- (CC-L methods L112-113): CC-L contains WMO-like cloud-type labels obtained by a fuzzy-logic classifier [6]. The classifier assigns one of eight classes that are comparable to the WMO cloud genera...;
- (Capabilities L328-333): The CCClim dataset presented in this paper allows for the investigation of clouds grouped by WMO-like cloud type with a long temporal coverage and high spatial resolution as daily samples. Using multiple cloud properties to define the classes makes them physically consistent and well-aligned with the morphological WMO genera. We showed that categorizing complex atmospheric data into established types is more expressive and interpretable than individual cloud properties;
- (Capabilities L337-338) This makes CCClim suitable for statistical analyses of clouds and enables quantification of seasonal cycles of WMO-like cloud types on a global scale.

**Other major comments:**

- Applying the ML algorithm cloud type to off-nadir MODIS pixels, when the training has been conducted with nadir MODIS cloud retrievals (that coincide with the active observations) can be justified only if it has been previously shown that the cloud retrievals are statistically the same at different parts of the MODIS swath, i.e., there are no biases in off-nadir retrievals (especially as one moves further aways from nadir).

While there is a difference in the retrievals of nadir and off-nadir cloud properties [2], quantifying its effect on cloud types is difficult, as there are no coinciding CC-L labels for off-nadir MODIS pixels. Discarding all off-nadir MODIS pixels would result in a considerable loss of spatial coverage. Since it seems reasonable to assume that differences in the retrieved physical cloud properties from nadir and off-nadir pixels are minimized by the MODIS retrievals (assuming the opposite would call into question the validity of much of the MODIS data), such viewing angle effects are expected to have a rather small effect on the results compared with other factors such as using a single representative cloud type for each column. This is now mentioned in L131-134.

-Similarly, applying the ML algorithm to ESACCI clouds (from AVHRR) has to be justified by showing that MODIS and AVHRR cloud property retrievals are statistically equivalent (the authors are aware they're not – lines 337-338). Because of different retrieval algorithms I doubt they are. Also, does ESACCI also include morning clouds? The training was conducted with afternoon clouds.

In [1] (Fig. 11, Table VI), the predictions of the ML algorithm from ESACCI data were compared to the distribution of the original CC-L labels, showing decent agreement for all cloud types. This comparison accounts for errors introduced by both stages of the ML algorithm and ensures that the *output* of the method is sensible even if the distribution of the *input* values in a different dataset is slightly different. This comparison is, therefore, more conclusive than a comparison of ESACCI and MODIS, as it shows that the method achieves the intended goal: reproducing the CC-L labels and being robust to small changes in the input data.

Different sampling of morning or afternoon clouds is not expected to influence the results significantly as the relationship between the physical cloud properties and the derived cloud type are essentially independent of the time of day. The incomplete sampling of the diurnal cycle could, however, influence the representation of daily mean values of the cloud-type RFOs. The RF is not very susceptible to small changes in the data due to a fairly high number of trees (400). We therefore do not expect the statistical differences mentioned by Referee #1 to have a discernible effect on the cloud-type distributions in CCClim.

– Similarly, applying the ML algorithm on climate models clouds is a huge stretch. What makes the model clouds equivalent to those of MODIS? You cannot even rigorously define a cloud-only grid column optical thickness when the grid is not overcast (depends on cloud fraction profile and overlap). At the minimum the model should've provided cloud output from the MODIS simulator.

We agree with the referee that care has to be taken when using CCClim to quantitatively evaluate a climate model. We, therefore, clarified in the revised version that we do not intend to do an actual evaluation of ICON-A, but only show an example of how CCClim could be used if climate model standard output would include variables like optical depth as instantaneous values. Using results obtained with a MODIS simulator in the model would certainly be desirable, but such a simulator is not readily available for ICON. We added the recommendation to use a satellite simulator (preferably Cloud\_cci) when using CCClim to evaluate climate models to Section 4. L324-326 now state that to perform actual climate model evaluation, instrument simulator output would be preferable.

Inconsistencies between the optical thickness values from ICON-A and the training data are, however, not expected to have a large impact as the latter contains grid-cell averages of the optical thickness, which is similar to the approach of ICON-A. The radiation parametrization of ICON-A uses the column-integrated water, such that computation of the column's optical thickness (in contrast to the resulting radiative fluxes) does not depend on the overlap assumption. We now make clear in L287-289 that our values are free from overlap assumptions.

Some minor comments:

– Figure 1 would have been more complete if the datasets used in each step (CUMULO, ESACCI, etc) were added as labels.

We updated Figure 1 to make it more clear what constitutes CUMULO. The ESACCI data is not included in Figure 1.

– Figure 2: Unclear to me: So if I properly weighted land and ocean and then normalized by the undetermined, I'd get close to what is shown in Fig. 2a? Perhaps you can say that.

This is correct, and we added this information to the figure caption.

- Figure 4: Why do this over the southern oceans where (beyond the nearly non-existent St) there is virtually no Cu and no DC? Perhaps show time series and trends for different regions for each cloud type, namely the areas where each dominates relative to its global mean? (Fig. 6)

Such an analysis would certainly be interesting but would in our opinion require at least four new plots equivalent to Figure 4, to illustrate the cloud-type behavior in selected regions of interest. We think this would be beyond the scope of a dataset description paper and rather belong into a scientific application paper.

- Figure 7: An interesting complement to Fig. 7 would be a table showing the contribution of each cloud type to the global CRE, weighted by RFO (subject to the disclaimer of no "pure" grid cells).

Unfortunately, such a detailed decomposition is not feasible due to the way the data have been processed. Simply weighting the results by RFO would assume that the total CRE can be linearly decomposed by cloud type, while in reality this is not the case.

– Lines 319-322: Of course there'll be few St in CCClim since you started with small frequency of St in CC-L. Same with DC. The difference between DC and St is though that DC are truly very rare while St are artificially rare in CC-L.

We think these limitations are noteworthy and would thus like to keep this statement in the text.

- Lines 344-346: Not sure what you mean by this.

We rephrased these sentences to clarify the meaning as follows (L370-376): The few datasets attempting

to assign objective WMO-like cloud types often disagree on important details. For example, while [4] are in good qualitative agreement with the ISCCP cloud distributions, they find, for instance, a much higher cirrus fraction in the tropics. In comparison, CCClim is less affected by problems common to these purely passive-sensor-based datasets, such as dealing with multilayer clouds [3]. Cluster-based datasets are however not suitable for a direct comparison with CCClim, as even intercomparison of datasets produced with unsupervised methods is difficult since the derived clusters do not have a common physical basis across all datasets.

**Referee: Peter Kuma**

We would also like to thank Peter Kuma for helping us to improve the manuscript.

Calling the CC-L cloud types WMO cloud types is misleading. The WMO cloud genera are defined using ground-based visual observations, as standardised by the WMO International Cloud Atlas. While CC-L definitions of cloud types aim to be similar to the WMO cloud genera, they are not the same. The deep convective (Dc) type is also not a WMO cloud genus. I think that the authors should be more clear about this.

We agree that calling the CC-L cloud types WMO cloud types is misleading and needs to be changed. We clarified the differences and changed the text to "WMO-like" throughout the manuscript. For a detailed answer and all changes applied to the revised version, please see our reply to comment (2) of referee #1 above.

There is a considerable overlap between the presented work and Kaps et al. (2023a). It is not entirely clear if the authors use the exact same methods as in the previous paper, or if anything is different. I think that the authors do not necessarily have to duplicate the description of the methods in this manuscript, and can instead reference the previous paper, which is really a pre-requisite for understanding this manuscript.

We clarified the overlap by saying explicitly (L149) that we use exactly the same RF model as the one trained for [1].

L177-178, Fig. 4 caption: The Cloud\_cci AVHRR L3U PM daily data are a composite of instantaneous values and not a daily mean, because the satellite overpass time is always at about the same local time, and the data have not been corrected with a diurnal cycle model. Stengel et al. (2020): 'All data are collected on two processing levels: (a) Level-3U, which represents daily composites of non-averaged data collected on a global latitude-longitude grid with 0.05° resolution and'. Unless some more sophisticated processing is done on the input data, the CCClim dataset also represents daily composites of instantaneous values.

There is some processing being done by averaging the RF outputs and the daily composite inputs, resulting in one mean value per day, which is (as the referee pointed out) not to be confused with a daily mean value averaging the whole diurnal cycle. This is now stated more clearly in L189.

It is not clear whether the results only represent daytime clouds or nighttime as well. The authors say that the Cumulo dataset is only available for daytime, but they do not say whether they apply the RF on nighttime grid cells from Cloud\_cci and ICON-A. If they only apply it on daytime, it should be made more clear that the results can be biased toward daytime clouds and not representative of polar night conditions. If they apply it on nighttime too, this should be mentioned as a limitation due to the fact that this is assuming that the statistical links between the input physical quantities and cloud types applying during daytime also apply during nighttime (reference training data for nighttime are lacking in this case).

The referee is correct that the RF is trained only on daytime data. ESACCI includes both, daytime and nighttime measurements, with nighttime data typically having higher uncertainties. We are not aware of reasons why the underlying statistical relationship between the retrieved physical cloud properties and the cloud types should be different during day and nighttime. A higher uncertainty in the nighttime retrievals is, however, expected to result in higher uncertainties in the derived cloud type. We see, for instance, a roughly 30% difference in the cloud/liquid/ice water path and optical depth retrievals between night and day. Therefore, we added to Section 5 (L364-365) that such deviations/uncertainties in the retrievals will have a corresponding effect on the CCClim cloud types.

Because the comparison between cloud types derived from ICON-A and Cloud\_cci is between 3-hourly instantaneous output of the model and daily composites of AVHRR measurements, it is not necessarily comparing the same time of day at local time. Because clouds tend to exhibit a diurnal cycle, this can introduce biases. This should be at least mentioned as a limitation. Also comparing 35 years of CCClim from AVHRR with 2 years from ICON-A is not necessarily a good idea because the global climate has changed during the last 35 years. Comparing the same 2 years of both would be better.

We agree with the points raised by the referee. Our comparison to ICON is only intended as an example and is not suitable to perform a quantitative model evaluation. This is now stated more clearly in the manuscript by naming these limitations and adding the recommendation to use a satellite simulator to create model output for evaluation to reduce biases introduced by the spatially and temporally incomplete sampling of the satellite as well as by instrument specific limitations. We also added the note that ideally either the same time periods from the model and the satellite data or climatological averages (20 years or more) should be compared. In order to illustrate the example application of CCClim for model evaluation, however, we do not think that additionally showing CCClim data only for the two ICON years would bring more insights.

One potential limitation that the authors do not mention is that cloud-related quantities such as cloud top pressure and cloud optical depth are not always directly comparable with those derived from radiance measured by a satellite instrument. For this reason, simulators for ISCCP and MODIS such as COSP exist. Because the RF takes these quantities as a input from the observations and the model without using an instrument simulator, an artificial bias can be introduced in the comparison.

We agree that using a satellite simulator to create the model output used for comparison would be ideal. For ICON-A, however, such output was not available. The example comparison shown here is only an illustration of how we envision CCClim to be used for model evaluation if the required output from climate models was available. We actually hope to encourage modelers with this example to provide such output in future simulations or model intercomparison projects that could then be used for a quantitative model evaluation. This is now stated more clearly at the end of Section 4.

The authors should include short sections in the Methods describing the Cumulo dataset and ICON-A. The section on CC-L should at least briefly describe how the CloudSat-CALIPSO cloud types are defined in the original dataset.

Following the referee's recommendation, we extended the MODIS and CC-L section to express more clearly how they constitute CUMULO (L106 and L113, respectively). As the main focus of the paper is to introduce CCClim and show some potential scientific applications, we think a more detailed model description beyond the references given and our description of the data we use is probably not adding much as we could have taken any model as an example with at least the basic output required for this illustration available.

A global regular angular (longitude-latitude) grid has unequal area of grid cells by latitude. This means that grid-cell averages are calculated over larger areas over polar latitudes than equatorial latitudes. This can have an effect on a predictor, such as the RF used on the coarse-grained Cloud\_cci data or ICON-A data. I think that this should be at least discussed.

As shown in [1], we do not expect this to be a big problem. We added this statement to the revised version (L152-153).

I think it would be possible to use a single machine learning model to determine the cloud types instead of two (IResNet on the fine scale followed by RF on the coarse scale). One could train a CNN to predict coarse grid cell cloud type fractions by training on grid cells where CC-L data are available (averaged over the coarse grid cell and represented as fractions of cloud types). It seems to me that the two step approach can be both more complex and less stable. Can the authors comment on this?

The suggested single-step approach would combine all of the reduction in resolution into one step, meaning that only a few labeled samples per cell would be considered representative of a complete grid cell as the CC-L data are very sparse (see e.g. the globes in Figure 1). Our 2-step approach ensures that cells without CC-L coverage adequately contribute to the cloud-type fractional amounts. One could argue that the second step is actually unnecessary for high-resolution applications, but as we show in [1], the intermediate step of pixel-wise labeling is crucial to take advantage of the active sensor data.

I suggest that the authors use terms 'cloud type' and 'cloud class' consistently throughout the manuscript. They seem to be used interchangeably, but it would be better to use either only one of them, or define the distinction between them.

We now consistently use "cloud type" throughout the text.

The active satellites sensors used as a reference for the CC-L cloud types have known problems with detecting low clouds. The CloudSat radar is affected by ground clutter, and the CALIPSO lidar signal is often attenuated by overlapping clouds before reaching mid and low-level clouds. These problems in turn also affect the presented dataset, and should be mentioned as a limitation.

Following the recommendation of the referee, we added these limitations to the text referring to Marchand et al. (2008) as a reference for CloudSat's difficulty in detecting low-level clouds due to surface clutter as an

example (L340-342). We would like to note that uncertainties in the CC-L cloud types resulting from these limitations are not mentioned in the CC-L ATBD [6].

The authors train the RF on MODIS but apply it to Cloud\_cci. It is not clear if this produces any sideeffects. Ideally, the authors should compare the results of applying the RF on MODIS and Cloud\_cci for the same testing time period (excluded from training).

As we could show in [1] by comparing the statistics of the predictions on ESACCI to the label distributions from the original CC-L dataset, application of the RF to ESACCI works quite well. A direct comparison of predictions from individual MODIS and ESACCI scenes as suggested by the referee would be valuable. But as the two datasets cannot easily be collocated, only a statistical comparison would be possible. In this respect, a statistical comparison with the CC-L ground truth is preferable to assess possible side-effects of the change in dataset.

Fig. 4: Long-term AVHRR datasets are usually not very well suited for determining trends due to discontinuities in the orbital parameters of the satellite series. This limitation can also translate to any derived dataset. Can the authors comment on whether the slopes shown in the figure are reliable? The authors mention this on L339, but it would be better to also caution the readers in the context of Fig. 4, and also on L8-10.

The CC4CL algorithm [5] used to create the ESACCI dataset attempts to remove inconsistencies between individual sensors without removing any trends. While the computed slopes should be treated with caution, though as mentioned in L339 of the initial submission. We added a disclaimer to the caption of Figure 4 that datasets derived from ESACCI, such as CCClim are not suitable for dedicated trend analyses. As trends are not mentioned in the abstract we would to keep the sentence in L8-10 as is.

L14-15: or its radiative effects: This could be a problem with reanalyses which usually parametrise clouds, which can result in radiation fluxes not corresponding to the actual cloud cover.

We rephrased this sentence to make it less ambiguous by replacing "or its radiative effects" with "in association with its properties".

L275: This waveband corresponds closely to the absorbing band used for the MODIS retrievals of cod and cer.: But what about AVHRR? Since the comparison here is between cloud types derived from ICON-A and from AVHRR.

As we could show in [1], application to ESACCI is possible without discernible side effects, even though AVHRR does not have a detector for this waveband. Thus, the validity of our example to illustrate an application of CCClim is not expected to be strongly affected. In a comprehensive climate model evaluation, though, one might want to investigate the effect of changing the waveband or preferably use an instrument simulator to create the model output for comparison. We added the recommendation to use a satellite simulator for such a comparison (see above for details).

L290-291: With respect to CCClim the simulation also exhibits an increase in high cloud RFO (Ci, Ns): Ns is classified as a middle level cloud in the WMO International Cloud Atlas.

Thanks for pointing this our. This was indeed misleading. We now write "increase in the RFO of higher clouds with significant ice content".

L351-352: there is arguably no better way to obtain cloud observations from space than combining radar, lidar and passive instruments.: I think that this is merely stating the obvious. Since there are currently no other satellite instruments relevant to cloud measurements other than lidar, radar and passive instruments, combining them necessarily has to give the best opportunity to obtain cloud measurements. One could also consider other kinds of passive measurements such as those by geosynchronous satellites, multi-angle measurements (MISR), passive microwave, Doppler lidar (Aeolus) and passive instruments on deep space satellites (DSCOVR). Including those in a combined analysis could produce more accurate results. The fuzzy logic classifier used in CC-L could have many different alternatives, for example using unsupervised learning or by supervised learning using ground-based observations. I do not think that one could say that it is the best, but rather one of many options of how to classify clouds.

We agree that including even more datasets is likely to further improve the results. We rephrased the sentence as follows: "Even though subjective thresholds are being used in the fuzzy logic classifier of CC-L, combining the different cloud observations from space radar, lidar and from passive instruments is a clear advantage over classification methods that do not use active sensor data such as such as many unsupervised approaches. Using additionally other kinds of passive measurements could further improve the results." (L381-385).

Technical comments Greater attention is needed to make the plots accessible to people with colour blindness. For example, in Fig. 1, 2, 6 and 7 some of the cloud type colours could not be distinguished. I recommend testing this for example with KMag on Linux.

We agree with the referee that this is an important aspect of any publication. This is why we made efforts in the initial manuscript to make the figure accessible to everyone by using a color scale especially designed to be colorblind-friendly and by using a colorblindness simulator. In another attempt, we are now using a different scale that is (supposedly) very accessible and recommended at https://clauswilke.com/dataviz/. L355: This section should contain links to all of the datasets used, including Cumulo, ERA5, MODIS, 2BCLDCLASS-LIDAR, Cloud\_cci and ICON-A.

The ESACCI and ERA5 data sources have been moved from the acknowledgments to the data availability section, and a link for CUMULO has been included. Since we did not use CC-L and MODIS data other than the data provided by CUMULO, we do not believe that additional links to the (original but unused) datasets used should be included here. We did not provide a link for ICON-A data as this was used for an example application only and no scientific conclusions are drawn concerning ICON that would justify the upload of a large volume of ICON results.

L372: It should be made clear that the DOI is a DOI.

Thanks for spotting this. Has been changed accordingly.

Fig. 2: This is more of a style issue, but pie charts are usually not considered a good visualisation method because the proportions are hard to compare. Bar charts usually provide a much better experience. We replaced the pie charts with stacked bars.

Fig. 3, 8, 9: Ideally, all colour bars should have labels. Fig. 4: The plots should have y-axis labels. Fig. 5: Plots should have a y-axis label RFO.

We adapted the Figures accordingly.

Fig. 10: The plots are probably too small. I recommend dedicating more space to the figure. The axes should have better labels, i.e. not just the cloud type name, but also that it is RFO on the x-axis, and probability density on the y-axis.

The extra labels have been included, the white space reduced and the height increased.

**References**

- Arndt Kaps, Axel Lauer, Gustau Camps-Valls, Pierre Gentine, Luis Gomez-Chova, and Veronika Eyring. Machine-learned cloud classes from satellite data for process-oriented climate model evaluation. *IEEE Transactions on Geoscience and Remote Sensing*, 61:1–15, 2023.
- [2] Michael D. King, Steven Platnick, W. Paul Menzel, Steven A. Ackerman, and Paul A. Hubanks. Spatial and temporal distribution of clouds observed by MODIS onboard the terra and aqua satellites. *IEEE Transactions on Geoscience and Remote Sensing*, 51(7):3826–3852, jul 2013.
- [3] J. Li, J. Huang, K. Stamnes, T. Wang, Q. Lv, and H. Jin. A global survey of cloud overlap based on CALIPSO and CloudSat measurements. *Atmospheric Chemistry and Physics*, 15(1):519–536, jan 2015.
- [4] C. J. Stubenrauch, A. Chédin, G. RÀdel, N. A. Scott, and S. Serrar. Cloud properties and their seasonal and diurnal variability from TOVS path-b. *Journal of Climate*, 19(21):5531–5553, nov 2006.
- [5] Oliver Sus, Martin Stengel, Stefan Stapelberg, Gregory McGarragh, Caroline Poulsen, Adam C. Povey, Cornelia Schlundt, Gareth Thomas, Matthew Christensen, Simon Proud, Matthias Jerg, Roy Grainger, and Rainer Hollmann. The community cloud retrieval for CLimate (CC4cl) – part 1: A framework applied to multiple satellite imaging sensors. Atmospheric Measurement Techniques, 11(6):3373–3396, jun 2018.
- [6] Zhien Wang. CloudSat 2B-CLDCLASS-LIDAR Product ProcessDescription and Interface Control Document, 2019.
- [7] WMO. International cloud atlas.

---

## Referee Report (RR1)

**Review of the revised manuscript 'Characterizing clouds with the CCClim dataset, a machine learning cloud class climatology'**

Peter Kuma

Department of Meteorology (MISU), Stockholm University, Stockholm SE-106 91, Sweden

6 February 2024

Dear editor and authors,

I would like to thank the authors for addressing my comments. I have a few remaining small comments, after which I recommend the manuscript for publication.

Kind regards,

Dr. Peter Kuma

**Comments**

L88: 'While these classes are defined to be consistent with the WMO definitions, misclassifications can occur, caused for example by the small footprint of the active sensors.': I think this is still overstating the consistency between the CC-L classes and WMO genera. The CC-L classes are defined based on a set of relatively synthetic (rule-based or fuzzy logic) thresholds. They are not expected to be matching statistically when compared to ground-based observations of clouds which are defined relatively vaguely based on a set of features determined visually by a person. Therefore, misclassifications are expected purely because the definition and the viewpoint are not the same. Also on L327: 'Using multiple cloud properties to define the classes makes them physically consistent and well-aligned with the morphological WMO genera.' And on L114: 'comparable': I am not sure if this is a good term to use. It would suggest that one could for example compare the CC-L RFOs with WMO cloud genera RFOs from a ground station and expect them to have comparable statistics. But that is not the case because the definition is different. It might be better to say 'analogous' or 'similar'.

One reason why training on daytime samples and applying the algorithm on nighttime might lead to biases is because nighttime passive retrievals are based on the infrared spectrum bands only, and lack any information provided by the visible spectrum bands. Therefore, there could be limitations on the minimal detectable cloud optical thickness and so on, and the results could have statistical biases compared to the daytime retrievals. The Cloud_cci nighttime products are considered of experimental quality, and they use different thresholds for day, night and twilight, according to Stengel et al., 2020: 'Night-time COT and CER retrievals are considered to be experimental products and only included in Level-3U products.', 'Please note that retrievals of CER, COT, CWP and CLA are also provided during night-time, although as experimental products.'.

L106: 'high temporal resolution': It is not clear what high resolution means. It is better to be more specific.

Fig. 4: 'daily mean RFO': As mentioned previously, I think that this is misleading because it is not representative of a daily temporal mean, but rather is a daily composite with an incomplete diurnal coverage.

Fig. 7: It might be good (but not necessary) to mention in the caption that the dashed line is SW CRE = LW CRE. 'Latitude Range: 0°N/S-90°N/S' is a bit unclear to me, but I guess it simply means global (?).

---

## Referee Report (RR2)

Review comments for "characterizing clouds with the CCClim dataset, a machine learning cloud class climatology" by Kaps et al.

This manuscript describes a cloud class climatology dataset that merges passive and active observations together in a fixed coarsened grid that is suitable for model comparisons. This dataset employs the CloudSat+CALIPSO 8 WMO cloud class as the "truth", and Aqua-MODIS Level 2 physical cloud property retrievals as the inputs for the training in consideration of obtaining large amount of collocated training samples and to make the model interpretable (hence not using MODIS Level 1 measurements). Then the labels predicted from native resolution MODIS data are then coarsened to GCM grid size and training coarse-grained MODIS data to generate another ML model that is finally applied to the ESACCI-AVHRR dataset to generate a 34-year long cloud class dataset. The authors then gave several examples illustrating how to use this dataset for GCM comparison to help identifying issues associated with certain cloud types.

Overall, I think there's some scientific merits of this current research. I particularly like the example regarding comparison with ICON-A outputs, and appreciate the "MODIS-equivalent filter" is applied to make the effort for a true apple-to-apple comparison. However, there are some fundamental issues with the design of the ML architecture that cause inevitable flaws to make this dataset really useful. These issues need to be addressed before publication of this work.

Major issues:

(1) The training "truth" CC-L coming from 2B-CLDCLASS only has CloudSat and some MODIS information. If you read the ATBD on CloudSat website, you'll find CALIPSO data is in their plan to use jointly, but was not implemented so far. The 2B-CLDCLASS-LIDAR product has cloud mask from joint CloudSat-CALIPSO observations, however, that dataset doesn't have cloud type classification. Because only CloudSat and limited MODIS information was fed into the "truth", it inherently underestimates cirrus clouds and then propagates this bias into your product.

(2) Aqua-MODIS always collocates with CloudSat at "nadir" view. That means the off-nadir correction must be made in order to not "overpredicting" cloud masks because of the slantwise integration length making the off-nadir view easier to detect cloud. This correction factor was never discussed in the current manuscript. The overprediction of the overall cloudiness in your CCClim dataset might likely attributes largely to this factor.

(3) For ancillary data from ERA5 reanalysis, I don't understand why temperature and water vapor profiles are not included. Aren't they the closest atmospheric variables to determine whether to form a cloud or not?

(4) It is not explained why you can apply two-step ML models trained by MODIS data directly to AVHRR data. Admittedly you use similar L2 cloud retrieval products for training and prediction, but AVHRR has so few bands (literally only one visible band), so the products are not quite comparable. Even the Cloud_CCI project had published a paper illustrating their discrepancies (https://doi.org/10.5194/essd-9-881-2017).

(5) By training on multi-year collocated MODIS-CloudSat data, I don't quite understand why the inter-annual variability is not learnt by the ML model, resulting in no-interannual variation (e.g., ENSO) in your timeseries shown in Fig. 4. Although it is clarified later on the manuscript that this dataset is not suitable for trend study, it is never claimed that it is not suitable for interannual variability study either. Ultimately, if a 34-yr long dataset is not intended for studying inter-annual variability, why product that? Why not just stop at Step-2 model and produce a MODIS cloud class that suits every application presented in this manuscript.

(6) As also notified in this manuscript, MODIS (all passive sensors) have issues distinguishing clouds against snow-cover surface in polar regions. However, the statistics (e.g., Fig. 2, Fig. 5) were summarized globally. I'd strongly suggest you exclude polar areas in computing your statistics.

Minor issues:

Fig. 3 – recommend adding a map from CCL for straightforward visual comparison. For example, I don't see the Gill model distribution in the Western Pacific cirrus clouds (might be obscured by the annual cycle or your coarse colorbar). Same recommendation for Fig. 5 (i.e., if using CCL annual cycle, do you still see the same bias suggested by your CCClim product?)

Fig. 10 – It's not understandable for cirrus clouds, how can ICON model produces a bunch of thin cirrus (i.e., IWP low) with large size ice particles? Please double check your graphing codes.

Line 277 – Eliasson et al., citation year missing.

---

## Author Response (AR2)

**Response to the Reviews**

April 9, 2024

Arndt Kaps
Deutsches Zentrum für Luft- und Raumfahrt e.V. (DLR)
Institut für Physik der Atmosphäre, Oberpfaffenhofen, Germany
Email: arndt.kaps@dlr.de

Earth System Science Data
Manuscript Number: essd-2023-424

Dear Editor: thank you for the opportunity to improve our manuscript. We have addressed the reviewer's comments below.

Yours sincerely,
*Arndt Kaps on behalf of the author team*

**Response to Referees**

**Earth System Science Data**

| | |
|---|---|
| Manuscript: | essd-2023-424 |
| Title: | Characterizing clouds with the CCClim dataset, a machine learning cloud class climatology |
| Authors: | Arndt Kaps, Axel Lauer, Rèmi Kazeroni, Martin Stengel, Veronika Eyring |
| Date: | April 9, 2024 |

We thank the two referees for their helpful comments. In this document, we answer each point raised by the referees. The original referees' comments are given in **blue**, our answers in **black**. All line numbers refer to the "track changes" version of the revised manuscript.

**Referee: 3**

Review comments for "characterizing clouds with the CCClim dataset, a machine learning cloud class climatology" by Kaps et al. This manuscript describes a cloud class climatology dataset that merges passive and active observations together in a fixed coarsened grid that is suitable for model comparisons. This dataset employs the CloudSat+CALIPSO 8 WMO cloud class as the "truth", and Aqua-MODIS Level 2 physical cloud property retrievals as the inputs for the training in consideration of obtaining large amount of collocated training samples and to make the model interpretable (hence not using MODIS Level 1 measurements). Then the labels predicted from native resolution MODIS data are then coarsened to GCM grid size and training coarse-grained MODIS data to generate another ML model that is finally applied to the ESACCI-AVHRR dataset to generate a 34-year long cloud class dataset. The authors then gave several examples illustrating how to use this dataset for GCM comparison to help identifying issues associated with certain cloud types. Overall, I think there's some scientific merits of this current research. I particularly like the example regarding comparison with ICON-A outputs, and appreciate the "MODISequivalent filter" is applied to make the effort for a true apple-to-apple comparison. However, there are some fundamental issues with the design of the ML architecture that cause inevitable flaws to make this dataset really useful. These issues need to be addressed before publication of this work.

Major issues: (1) The training "truth" CC-L coming from 2B-CLDCLASS only has CloudSat and some MODIS information. If you read the ATBD on CloudSat website, you'll find CALIPSO data is in their plan to use jointly, but was not implemented so far. The 2B-CLDCLASS-LIDAR product has cloud mask from joint CloudSat-CALIPSO observations, however, that dataset doesn't have cloud type classification. Because only CloudSat and limited MODIS information was fed into the "truth", it inherently underestimates cirrus clouds and then propagates this bias into your product.

According to the CloudSat Product Process Description and Interface Control Document (PDICD [1]), the 2B-CLDCLASS-LIDAR dataset used here provides cloud type information based on both, CPR and CALIOP data. We agree with the reviewer that the radar-only dataset 2B-CLDCLASS does not contain this information, but this dataset is not used here.

(2) Aqua-MODIS always collocates with CloudSat at "nadir" view. That means the off-nadir correction must be made in order to not "overpredicting" cloud masks because of the slantwise integration length making the off-nadir view easier to detect cloud. This correction factor was never discussed in the current manuscript. The overprediction of the overall cloudiness in your CCClim dataset might likely attributes largely to this factor.

When creating CCClim, we followed the approach typically taken in existing literature regarding nadir/off-nadir effects, which is to prefer coverage over precision. But the reviewer has a good point.

In order to estimate possible effects of the viewing angle on the cloud classes in CCClim, we created a second version of the dataset by discarding pixels that are presumably affected by large viewing-angle-related uncertainties. For this, we removed the 300 outermost pixels on each side when training the Random Forest (i.e. in each across-track line 600 of 1354 pixels are removed). As expected, this leads to an increase in the
* * *
[1] https://www.cloudsat.cira.colostate.edu/cloudsat-static/info/dl/2b-cldclass-lidar/2B-CLDCLASS-LIDAR_PDICD.P1_R05.rev0_.pdf

fraction of "undetermined" pixels per training grid cell, causing a reduction in performance for the other eight classes as visible in their mean $R^2$-Score of $R^2_{cropped} = 0.79$, compared to the original score of $R^2_{orig} = 0.84$, despite applying weights in their favor (see Kaps et al., 2023). We then performed a comparison equivalent to Table VI in Kaps et al. (2023), which assesses how well CCClim captures the average global distribution of cloud types as in CC-L. For every cloud type, the correlation with CC-L is much lower than for the original CCClim. This suggests that the inclusion of all available training data, even though they are affected by viewing angle effects to some degree, is beneficial overall.

A correction of the viewing angle effects as proposed by the reviewer depends on cloud type and cannot be easily done since scattering angle and solar zenith angle are not trivially related for each specific pixel (Maddux et al., 2010; Painemal et al., 2021). There are no correction methods proposed in literature for the cloud property retrievals. Examining a possible correction method would require re-deriving the level-2 MODIS products (Bennartz and Rausch, 2017) and is therefore beyond the scope of this paper.

In the previous version of the manuscript we mention that CCClim is produced under the assumption that off-nadir retrievals are of similar quality than the nadir-near retrievals. The point raised by the reviewer made it clear that this statement is oversimplified. We therefore added the following paragraphs to the revised manuscript:

- (lines 134-140) While viewing geometry is part of the retrievals for the cloud optical properties (Platnick et al., 2017), across-track angular dependencies have been identified as a source of uncertainties in the MODIS Collection 5 data (Horvath et al., 2014; Maddux et al., 2010). Similar effects are expected for the Collection 6 data used here, but their magnitude is unclear and no published correction methods are available. These effects are, however, expected to be less relevant for long-term averages (Maddux et al., 2010). Following the data usage in many other studies (e.g. Bennartz and Rausch, 2017; Cho et al., 2021; Oreopoulos et al., 2016) we have therefore opted to make the above assumption instead of introducing new uncertainties by trying to correct these non-trivial effects.

- (lines 151-155) Sensitivity tests with the outermost 300 pixels on each side discarded during training of the RF to reduce the viewing angle effects (not shown) show a decreased performance for all of the eight cloud classes as measured by their mean $R^2$-Score and a smaller correlation with the CC-L ground-truth. We therefore used the complete breadth of the swath to generate CCClim despite the quality degradation towards the edges of the swath.

We furthermore now mention explicitly that viewing angle effects introduce additional uncertainties in CCClim by adding lines 351-356:

- Also, since CC-L labels are only available for nadir-near MODIS pixels, uncertainties might be introduced by labeling off-nadir pixels in the classification stage. A correction, however, is non-trivial (e.g. Painemal et al., 2021) and no concrete solutions are available in literature. Simply discarding pixels that are "far" away from the center reduces the data coverage and results in a significant reduction in performance. Even though the viewing angle effect is very hard to quantify, it is expected to be at least somewhat mitigated through the spatial and temporal aggregation in CCClim (e.g. Bennartz and Rausch, 2017; Maddux et al., 2010).

(3) For ancillary data from ERA5 reanalysis, I don't understand why temperature and water vapor profiles are not included. Aren't they the closest atmospheric variables to determine whether to form a cloud or not? The ERA5 data used in this work are only an example of how CCClim can be complemented by additional data for further analysis. No ERA5 data are provided as ancillary variables with CCClim. The example is meant to show that the CCClim cloud types hold relevant information comparable to that of other techniques when categorizing clouds by dynamical regime using $\omega_{500}$ as a proxy as done by Bony et al. (2004). This classification of the dynamical regime does not rely on other parameters such as water vapor or temperature profiles. Additionally, these variables are not particularly informative for the cloud *type*, but more for a cloud mask, as indicated by the reviewer.

(4) It is not explained why you can apply two-step ML models trained by MODIS data directly to AVHRR data. Admittedly you use similar L2 cloud retrieval products for training and prediction, but AVHRR has so

[Figure]

Figure 1: Relative deviations from average annual cycle (blue) averaged over the region $(30, 60)°N, (60, 0)°W$ with rolling average (red) showing the interannual variability.

few bands (literally only one visible band), so the products are not quite comparable. Even the Cloud_CCI project had published a paper illustrating their discrepancies (https://doi.org/10.5194/essd-9-881-2017).
The paper mentioned by the reviewer illustrates that the two datasets deviate in particular in their overall/liquid cloud fraction, which are not included in the values used by us. We showed in (Kaps et al., 2023) that predictions obtained from AVHRR data with our method are in good statistical agreement with the CC-L ground truth. This suggests that the differences between MODIS and AVHRR do not seem to play a major role. Being able to apply the ML-model to independent other datasets is a prerequisite for application to climate model data for model evaluation, an important aim of this ML framework. This is mentioned in lines 149-151.
(5) By training on multi-year collocated MODIS-CloudSat data, I don't quite understand why the interannual variability is not learnt by the ML model, resulting in no interannual variation (e.g., ENSO) in your timeseries shown in Fig. 4. Although it is clarified later on the manuscript that this dataset is not suitable for trend study, it is never claimed that it is not suitable for interannual variability study either. Ultimately, if a 34-yr long dataset is not intended for studying inter-annual variability, why product that? Why not just stop at Step-2 model and produce a MODIS cloud class that suits every application presented in this manuscript.

Interannual variability is learned by the ML model as we can see in our results. Interannual variability is, however, very hard to see in Fig. 4 as we show simply an average over all ocean grid cells in the Southern Hemisphere to illustrate the daily mean RFO and its standard deviation for different cloud types. For interannual variability studies the average seasonal cycle is typically subtracted from the time series and often smaller regions are investigated. As an example to show that interannual variability can indeed be studied with CCClim, we present a time series of the anomalies in RFO (i.e. the average seasonal cycle is subtracted) averaged over the north Atlantic $((30, 60)°N, (60, 0)°W)$, for the 3 years from Jan. 1982 to Dec. 1987. Figure 1 shows that the interannual variability of cloud types can be seen in CCClim with periods of positive and negative anomalies (visible for example in the anomaly for Ci).
(6) As also notified in this manuscript, MODIS (all passive sensors) have issues distinguishing clouds against snow-cover surface in polar regions. However, the statistics (e.g., Fig. 2, Fig. 5) were summarized globally. I'd strongly suggest you exclude polar areas in computing your statistics.
This is a sensible suggestion and for the example analyses we show, the samples are cleared of possibly ice-covered ocean, as is stated in the manuscript. However, this manuscript is intended to introduce the CCClim dataset and not meant as a scientific analysis. We therefore believe it is more helpful to show global statistics rather than masked values. We added lines 366-369 to inform users of this dataset of this potential error source.

Minor issues: Fig. 3 - recommend adding a map from CCL for straightforward visual comparison.
Such a figure is already shown in (Kaps et al., 2023) (their figure 8) and we would prefer to not repeat this figure here.
For example, I don't see the Gill model distribution in the Western Pacific cirrus clouds (might be obscured by the annual cycle or your coarse colorbar). Same recommendation for Fig. 5 (i.e., if using CCL annual cycle, do you still see the same bias suggested by your CCClim product?)
We think in order to study atmosphere-ocean interactions such as the ones mentioned by the reviewer, one would have to look at anomalies rather than the absolute values of the seasonal cycle. Such an analysis could possibly be done with CCClim, but is in our opinion beyond the scope of this paper that aims at introducing CCClim and providing a few examples of what could be done with the CCClim dataset. As the CC-L data are very sparse, we do not expect that a robust annual cycle can be calculated that would be needed for a meaningful comparison.
Fig. 10 - It's not understandable for cirrus clouds, how can ICON model produces a bunch of thin cirrus (i.e., IWP low) with large size ice particles? Please double check your graphing codes.
The parameterization of the effective radius of ice particles $r_{e,i}$ in the ICON-A version used here depends only on the in-cloud ice water content $q_i$ as there is no information on the ice crystal number concentration: $r_{e,i} = 83.8 \cdot q_i^{0.216}$. This means that the vertically integrated cloud ice water content shown in Fig. 10 (gridbox-averages, output of ICON-A) can be small if the cloud fraction is small and/or cloud ice is only present in one or two model layers. Yet, $r_{e,i}$ at cloud top shown in Figure 10 can have relatively large values that are typical for cirrus clouds.

**Referee: Peter Kuma**

We would also like to thank Peter Kuma for helping us to improve the manuscript.

L88: 'While these classes are defined to be consistent with the WMO definitions, misclassifications can occur, caused for example by the small footprint of the active sensors.': I think this is still overstating the consistency between the CC-L classes and WMO genera. The CC-L classes are defined based on a set of relatively synthetic (rule-based or fuzzy logic) thresholds. They are not expected to be matching statistically when compared to ground-based observations of clouds which are defined relatively vaguely based on a set of features determined visually by a person. Therefore, misclassifications are expected purely because the definition and the viewpoint are not the same.

The fuzzy logic algorithm is designed in a way that the physical features of these cloud genera are within the range of what is shown by observations of these cloud types over many decades (Wang, 2019). We agree that cloud types from CC-L and ground-based observations are not exactly the same because of the very different approach of both methods. Here, it helps that active sensors are sometimes able to determine also the cloud base height, which is expected to help with providing at least some consistency with the surface-based classifications. Following the point of the reviewer, we rephrased the corresponding statement to make clear that even though the CC-L cloud classes are not matching the WMO-classes, they have been designed in a way that the retrieved physical properties are within the range of typical values for these cloud classes.

Also on L327: 'Using multiple cloud properties to define the classes makes them physically consistent and well-aligned with the morphological WMO genera.' And on L114: 'comparable': I am not sure if this is a good term to use. It would suggest that one could for example compare the CC-L RFOs with WMO cloud genera RFOs from a ground station and expect them to have comparable statistics. But that is not the case because the definition is different. It might be better to say 'analogous' or 'similar'.

Following the suggestions of the reviewer, we rephrased the sentence using 'similar'.

One reason why training on daytime samples and applying the algorithm on nighttime might lead to biases is because nighttime passive retrievals are based on the infrared spectrum bands only, and lack any information provided by the visible spectrum bands. Therefore, there could be limitations on the minimal detectable cloud optical thickness and so on, and the results could have statistical biases compared to the daytime retrievals. The Cloud_cci nighttime products are considered of experimental quality, and they use different thresholds for day, night and twilight, according to Stengel et al., 2020: 'Night-time COT and CER retrievals are considered to be experimental products and only included in Level-3U products.', 'Please note that retrievals of CER, COT, CWP and CLA are also provided during night-time, although as experimental products.'.

We agree with the reviewer that the nighttime products are experimental and thus subject to larger uncertainty than their daytime counterparts. We emphasized this by explicitly mentioning in the manuscript (lines 379-380) that the nighttime retrievals are experimental.

L106: 'high temporal resolution': It is not clear what high resolution means. It is better to be more specific.

Changed to "daily resolution".

Fig. 4: 'dailymean RFO': As mentioned previously, I think that this is misleading because it is not representative of a daily temporal mean, but rather is a daily composite with an incomplete diurnal coverage.

Agreed, changed to "daily RFO values".

Fig. 7: It might be good (but not necessary) to mention in the caption that the dashed line is SW CRE = LW CRE. 'Latitude Range: $0°N/S - 90°N/S$' is a bit unclear to me, but I guess it simply means global (?)

In order to avoid confusion, we removed the title and added the dashed line to the legend.

**References**

Bennartz, R. and J. Rausch (2017). "Global and regional estimates of warm cloud droplet number concentration based on 13 years of AQUA-MODIS observations". In: *Atmospheric Chemistry and Physics* 17.16, pp. 9815–9836. ISSN: 1680-7324. DOI: 10.5194/acp-17-9815-2017.

Bony, S., J.-L. Dufresne, H. L. Treut, J.-J. Morcrette, and C. Senior (2004). "On dynamic and thermodynamic components of cloud changes". In: *Climate Dynamics* 22.2-3, pp. 71–86. DOI: 10.1007/s00382-003-0369-6.

Cho, N., J. Tan, and L. Oreopoulos (2021). "Classifying planetary cloudiness with an updated set of MODIS Cloud Regimes". In: *Journal of Applied Meteorology and Climatology*. DOI: 10.1175/jamc-d-20-0247.1.

Horvath, Ã., C. Seethala, and H. Deneke (2014). "View angle dependence of MODIS liquid water path retrievals in warm oceanic clouds". In: *Journal of Geophysical Research: Atmospheres* 119.13, pp. 8304–8328. ISSN: 2169-8996. DOI: 10.1002/2013jd021355.

Kaps, A., A. Lauer, G. Camps-Valls, P. Gentine, L. Gomez-Chova, and V. Eyring (2023). "Machine-Learned Cloud Classes From Satellite Data for Process-Oriented Climate Model Evaluation". In: *IEEE Transactions on Geoscience and Remote Sensing* 61, pp. 1–15. DOI: 10.1109/TGRS.2023.3237008.

Maddux, B. C., S. A. Ackerman, and S. Platnick (2010). "Viewing Geometry Dependencies in MODIS Cloud Products". In: *Journal of Atmospheric and Oceanic Technology* 27.9, pp. 1519–1528. ISSN: 0739-0572. DOI: 10.1175/2010jtecha1432.1.

Oreopoulos, L., N. Cho, D. Lee, and S. Kato (2016). "Radiative effects of global MODIS cloud regimes". In: *Journal of Geophysical Research: Atmospheres* 121.5, pp. 2299–2317. DOI: 10.1002/2015jd024502.

Painemal, D. et al. (2021). "Evaluation of satellite retrievals of liquid clouds from the GOES-13 imager and MODIS over the midlatitude North Atlantic during the NAAMES campaign". In: *Atmospheric Measurement Techniques* 14.10, pp. 6633–6646. ISSN: 1867-8548. DOI: 10.5194/amt-14-6633-2021.

Platnick, S. et al. (2017). "The MODIS Cloud Optical and Microphysical Products: Collection 6 Updates and Examples From Terra and Aqua". In: *IEEE Transactions on Geoscience and Remote Sensing* 55.1, pp. 502–525. DOI: 10.1109/tgrs.2016.2610522.

Wang, Z. (2019). *CloudSat 2B-CLDCLASS-LIDAR Product ProcessDescription and Interface Control Document*. Version p1_R05. URL: https://www.cloudsat.cira.colostate.edu/data-products/2b-cldclass-lidar.

---

## Author Response (AR3)

**Response to the Reviews**

May 2, 2024

Arndt Kaps
Deutsches Zentrum für Luft- und Raumfahrt e.V. (DLR)
Institut für Physik der Atmosphäre, Oberpfaffenhofen, Germany
Email: arndt.kaps@dlr.de

Earth System Science Data
Manuscript Number: essd-2023-424

Dear Editor:

We have taken the reviewer's comments into account and agree that application of ML models to out-of-distribution data such as a different sensor is generally very difficult and great care has to be taken. As will be elaborated below, our approach is justified, however, due to both the similarities of the datasets (intercalibration, heritage channels) and the fact that we found good validation results against the 2B-CLDCLASS-LIDAR (CC-L) ground truth.

The main points of this discussion have been added to the revised manuscript.

Yours sincerely,

*Arndt Kaps on behalf of the author team*

**Response to Referees**

**Earth System Science Data**

| | |
|---|---|
| Manuscript: | essd-2023-424 |
| Title: | Characterizing clouds with the CCClim dataset, a machine learning cloud class climatology |
| Authors: | Arndt Kaps, Axel Lauer, Rèmi Kazeroni, Martin Stengel, Veronika Eyring |
| Date: | May 2, 2024 |

**Referee: 3**

We thank the anonymous referee for his/her comments and for supporting publication of our manuscript. The original referee's comments are given in **blue**, our answers in **black**. All line numbers refer to the "track changes" version of the revised manuscript.

Overall I'm satisfied with your responses and revisions, but point #4 involves a fundamental issue for transfer learning that cannot be overlooked no matter viewing this issue from ML angle or from data rigor angle. Unfortunately this issue relates to the fundamental approach you use so I have to suggest major revision. Having said that, I do see and appreciate the discussion on scientific applications of your dataset. So I wholeheartedly support a final publication of your paper and dataset. ML is a great tool and should be embraced by our community, but we can't lose the rigor (which is also strongly suggested by numerous AI/ML research). In your case, it can be fixed using a collocated AVHRR-CC-L/CALIPSO dataset for training.

The second stage of the ML algorithm (random forest regression) is needed to be able to apply the algorithm to coarse-resolution data (having global climate models in mind). We share the concern of the reviewer that the ML algorithm might not be applicable to other datasets than MODIS. For this reason, we verified that the ML algorithm can indeed be applied to the out-of-distribution ESACCI dataset by comparing predictions of the ML algorithm applied to ESACCI data to the distribution of the original CC-L cloud class labels. As documented in Kaps et al. (2023), we found reasonable reproduction of the geographical distribution of all cloud types, which suggests that the method is robust enough to be applied to different input datasets if the data represent similar basic physical properties. In many settings, ML models have to be applied out-of-distribution an Kuma et al. (2023), Wang (2019), and Yuval and O'Gorman (2020) are some examples in climate science. When doing so, it is important to quantify the uncertainties induced by the domain shift, which for our method is documented in (Kaps et al., 2023).

Therefore, we disagree with the reviewer that in the case of deriving cloud class labels from physical cloud properties, the ML algorithm trained on MODIS data cannot be applied to AVHRR data. As an example, we reproduce Table VI from Kaps et al. (2023) quantifying the differences and correlations between the mean predicted fractions and CC-L per $2° \times 2°$ grid cell, thus demonstrating reasonable agreement 1 and allowing estimation of the uncertainty when applied to ESACCI.

We are using ESA's Cloud_cci dataset as it provides a consistent, long-term time series of cloud properties obtained from harmonized, reprocessed products from different satellite instruments. Especially relevant in

Table 1: Mean fraction of the predicted classes compared with the relative amounts of the classes in CC-L. $c_P$ is the Pearson-correlation between the geographical distributions. The last row shows the mean difference for pixels with predictions in the 90th percentile $\Delta_{90}$ relative to the mean $\mu_{90}$ of these predictions. Predictions are taken from a model trained on $(100 \text{ km})^2$ from CUMULO and applied on $100 \times 100$ pixel Cloud_cci grid cells.

| | Ci | As | Ac | St | Sc | Cu | Ns | Dc |
|---|---|---|---|---|---|---|---|---|
| Predict. | 0.13 | 0.14 | 0.19 | 0.01 | 0.31 | 0.10 | 0.10 | 0.02 |
| CloudSat | 0.20 | 0.13 | 0.11 | 0.05 | 0.27 | 0.12 | 0.09 | 0.03 |
| $c_P$ | 0.87 | 0.80 | 0.60 | 0.18 | 0.88 | 0.84 | 0.83 | 0.36 |
| $\Delta_{90}/\mu_{90}$ | -29% | 49% | 49% | 1% | 18% | 14% | 30% | 39% |

that case is that two of the five AVHRR channels are intercalibrated with MODIS Aqua (Heidinger et al., 2010; Stengel et al., 2017) and that MODIS uses AVHRR heritage channels for its cloud property retrievals (Platnick et al., 2017). Version 3 of Cloud_cci has been shown to provide good global quality scores for cloud detection, cloud phase and ice water path based on validation results against A-Train sensors (Stengel et al., 2020). Collocating Cloud_cci data with CC-L is, however, non-trivial and would only provide a small number of usable samples for the few instances in which orbits overlap. Therefore, this collocation is in our opinion beyond the scope of this study.

Following the point of the reviewer, we added more discussion on the out-of-distribution application of our ML algorithm to the revised manuscript (lines 93-100):
"Application of an ML model to a different dataset (out-of-distribution) can be problematic and great care has to be taken to ensure no unexpected errors occur. Out-of-distribution application of ML models has been done in climate science before(e.g Kuma et al., 2023; Wang, 2019; Yuval and O'Gorman, 2020). Here, it is important to estimate the uncertainties induced by the domain shift. In our case, the regression model trained on MODIS cloud properties is applied to similar AVHRR retrievals. Applicability could be demonstrated by the reasonable reproduction of the geographical distribution of all cloud types when applying the model to ESACCI data (AVHRR), also providing an uncertainty estimate as documented in Kaps et al. (2023). This suggests that the method is robust enough to be applied to different datasets if they represent similar basic physical properties."

**References**

Heidinger, A. K., W. C. Straka, C. C. Molling, J. T. Sullivan, and X. Wu (2010). "Deriving an inter-sensor consistent calibration for the AVHRR solar reflectance data record". In: *International Journal of Remote Sensing* 31.24, pp. 6493–6517. ISSN: 1366-5901. DOI: 10.1080/01431161.2010.496472.

Kaps, A., A. Lauer, G. Camps-Valls, P. Gentine, L. Gomez-Chova, and V. Eyring (2023). "Machine-Learned Cloud Classes From Satellite Data for Process-Oriented Climate Model Evaluation". In: *IEEE Transactions on Geoscience and Remote Sensing* 61, pp. 1–15. DOI: 10.1109/TGRS.2023.3237008.

Kuma, P., F. A.-M. Bender, A. Schuddeboom, A. J. McDonald, and Ø. Seland (2023). "Machine learning of cloud types in satellite observations and climate models". In: *Atmospheric Chemistry and Physics* 23.1, pp. 523–549. DOI: 10.5194/acp-23-523-2023.

Platnick, S. et al. (2017). "The MODIS Cloud Optical and Microphysical Products: Collection 6 Updates and Examples From Terra and Aqua". In: *IEEE Transactions on Geoscience and Remote Sensing* 55.1, pp. 502–525. DOI: 10.1109/tgrs.2016.2610522.

Stengel, M. et al. (2017). "Cloud property datasets retrieved from AVHRR, MODIS, AATSR and MERIS in the framework of the Cloud_cci project". In: *Earth System Science Data* 9.2, pp. 881–904. DOI: 10.5194/essd-9-881-2017.

Stengel, M. et al. (2020). "Cloud_cci Advanced Very High Resolution Radiometer post meridiem (AVHRR-PM) dataset version 3: 35-year climatology of global cloud and radiation properties". In: *Earth System Science Data* 12.1, pp. 41–60. DOI: 10.5194/essd-12-41-2020.

Wang, Z. (2019). *CloudSat 2B-CLDCLASS-LIDAR Product ProcessDescription and Interface Control Document.* Version p1_R05. URL: https://www.cloudsat.cira.colostate.edu/data-products/2b-cldclass-lidar.

Yuval, J. and P. A. O'Gorman (2020). "Stable machine-learning parameterization of subgrid processes for climate modeling at a range of resolutions". In: *Nature Communications* 11.1. ISSN: 2041-1723. DOI: 10.1038/s41467-020-17142-3.